# UaiNets: From Unsupervised to Active Deep Anomaly Detection

## Abstract

This work presents a method for *active anomaly detection* which can be built upon existing deep learning solutions for unsupervised anomaly detection. We show that a prior needs to be assumed on what the anomalies are, in order to have performance guarantees in unsupervised anomaly detection. We argue that active anomaly detection has, in practice, the same cost of unsupervised anomaly detection but with the possibility of much better results. To solve this problem, we present a new layer that can be attached to any deep learning model designed for unsupervised anomaly detection to transform it into an active method, presenting results on both synthetic and real anomaly detection datasets.

## 1 Introduction

Anomaly detection (a.k.a. outlier detection) (Hodge & Austin, 2004; Chandola et al., 2009; Aggarwal, 2015) aims to discover rare instances that do not conform to the patterns of majority. From a business perspective, though, we are not only interested in finding rare instances, but "usefull anomalies". This problem has been amply studied recently (Liu et al., 2017; Li et al., 2017; Zong et al., 2018; Maurus & Plant, 2017; Zheng et al., 2017), with solutions inspired by extreme value theory (Siffer et al., 2017), robust statistics (Zhou & Paffenroth, 2017) and graph theory (Perozzi et al., 2014).

Unsupervised anomaly detection is a sub-area of outlier detection, being frequently applied since label acquisition is very expensive and time consuming. It is a specially hard task, where there is usually no information on what these rare instances are and most works use models with implicit priors or heuristics to discover these anomalies, providing an anomaly score $s(x)$ for each instance in a dataset. Active anomaly detection is a powerful alternative approach to this problem, which has presented good results in recent works such as (Veeramachaneni et al., 2016; Das et al., 2016; 2017).

In this work, we first show that unsupervised anomaly detection requires priors to be assumed on the anomaly distribution; we then argue in favor of approaching it with active anomaly detection, an important, but under-explored approach (Section 2). We propose a new layer, called here Universal Anomaly Inference (UAI), which can be applied on top of any unsupervised anomaly detection model based on deep learning to transform it into an active model (Section 3). This layer uses the strongest assets of deep anomaly detection models, i.e. its learned latent representations ($l$) and anomaly score ($s$), to train a classifier on the few already labeled instances. An example of such an application can be seen in Figure 1, where an UAI layer is built upon a Deanoising AutoEncoder (DAE).

We then present extensive experiments, analyzing the performance of our systems vs unsupervised, semi-supervised and active ones under similar budgets in both synthetic and real data, showing our algorithm improves state of the art results in several datasets, with no hyperparameter tuning (Section 4). Finally, we visualize our models learned latent representations, comparing them to unsupervised models' ones and analyze our model's performance for different numbers of labels (Appendix C).

## 2 Problem Definition

Grubbs (1969) defines an outlying observation, or outlier, as one that appears to deviate markedly from other members of the sample in which it occurs. Hawkins (1980) states that an outlier is an observation that deviates so much from other observations as to arouse suspicion that it was generated by a different mechanism. While Chandola et al. (2009) says that normal data instances occur in high probability regions of a stochastic model, while anomalies occur in the low probability ones. Following these definitions, specially the one from (Hawkins, 1980), we assume there is a probability density function

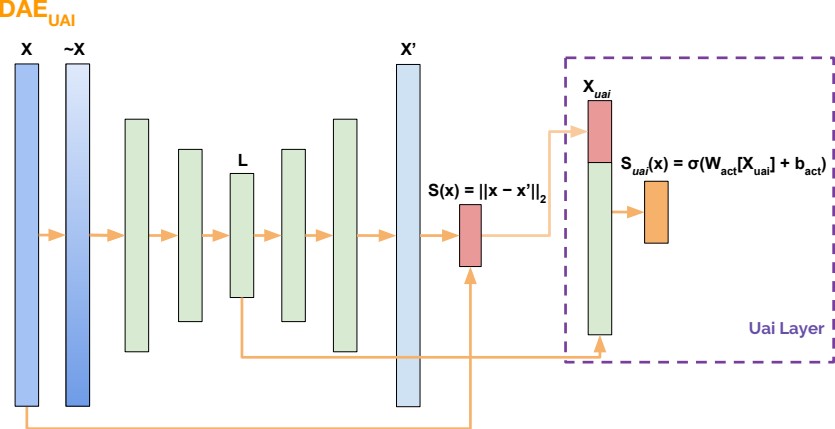

Figure 1: $DAE_{uai}$ architecture.

from which our 'normal' data instances are generated: $X_{normal} \sim p_{normal}(x) = p(x|y = 0)$, where $x$ is an instance's available information[1] and $y$ is a label saying if the point is anomalous or not. There is also a different probability density function from which anomalous data instances are sampled: $X_{anom} \sim p_{anom}(x) = p(x|y = 1)$.

A full dataset is composed of both normal and anomalous instance, being sampled from a probability distribution that follows:

$$
\begin{array}{rclcl}
(X, Y)_{full} & \sim & p_{full}(x, y) & = & p(y)\, p(x|y) \\
X_{full} & \sim & p_{full}(x) & = & p(y = 0)\, p_{normal}(x) + p(y = 1)\, p_{anom}(x) \\
& & & = & (1 - \lambda) p_{normal}(x) + \lambda p_{anom}(x)
\end{array} \tag{1}
$$

where $\lambda$ is an usually small constant representing the probability of a random data point being anomalous ($\lambda = p(y = 1)$), this constant can be either known a priori or not. Chandola et al. (2009) divides anomaly detection learning systems in three different types:

- Supervised: A training and a test set are available with curated labels for non-anomalous and anomalous instances. This case is similar to an unbalanced supervised classification setting:
$$
\mathcal{D}_{train/test} = (X, Y)_{train/test} \sim p_{full}(x, y)
$$

- Semi-Supervised: A training set is available containing only non-anomalous instances and the challenge is to identify anomalous instances in a test set. This is also called novelty detection:
$$
\begin{array}{l}
\mathcal{D}_{train} = X_{train} \sim p_{normal}(x) \\
\mathcal{D}_{test} = X_{test} \sim p_{full}(x)
\end{array}
$$

- Unsupervised: A dataset containing both non-anomalous and anomalous instance is available and the challenge is to identify anomalous instances in it. There is no concept of a test set since anomalous instances must be sorted in the dataset itself:
$$
\mathcal{D} = X \sim p_{full}(x)
$$

## 2.1 UNSUPERVISED ANOMALY DETECTION

In this work, we will focus on *unsupervised anomaly detection*. Here, in possession of the full set of points $X \sim p_{full}(x)$, we want to find a subset $X_{anom} \subset X$ which is composed of the anomalous instances. The full distribution $p_{full}$ is a mixture of distributions and if these distributions overlap very closely, it may be impossible to learn the individual distributions beyond a certain accuracy threshold (Dasgupta et al., 2005). It is a well-known result that general mixture models are unidentifiable (Aragam et al., 2018; Bordes et al., 2006). In the sequence, we further show that we gain no information on $p_{anom}$ from $p_{full}$ for any small $\lambda$ without a prior on the anomalies' probability distribution. This differs from the usual unidentifiability of mixture models result in that we make no assumptions on the prior for $p_{normal}$, showing all valid distributions of $p_{anom}$ are equally probable.

---

[1]$x$, in our notation, is the information known about a data instance. This can be further composed of what would actually be $x$ and $y$ in a supervised setting, such as an image and its corresponding class label. We will reference this as $x_x$ and $x_y$ here.

**Theorem 1.** *No free anomaly theorem. Consider two independent arbitrary probability distributions* $p_{normal}$ *and* $p_{anom}$. *For a small number of anomalies* $\lambda \approx 0$, $p_{full} = \bar{p}$ *gives us no further knowledge on the distribution of* $p_{anom}$:

$$p(p_{anom}|p_{full} = \bar{p}) \approx Uniform(P_2), \quad \lambda \approx 0$$

*where*

$$\begin{aligned} P_2 &= \{p_r, \forall p_r \in P \mid \lambda \in [0;1], \lambda \cdot p_r \leq \bar{p}\} \\ \lim_{\lambda \to 0} P_2 &= \{p, \forall p \in P \mid supp(p) \subseteq supp(\bar{p})\} \end{aligned}$$

From Theorem 1 we can conclude that unsupervised anomaly detection requires a prior on the anomalies distribution. A more tangible example of this can be seen in Figure 2, where we present a synthetic data distribution composed of three classes of data clustered in four visibly separable clusters. Anomaly detection is an undecidable problem in this setting without further information, since it is impossible to know if the low density cluster is composed of anomalies or the anomalies are the unclustered low density points (or a combination of both).

If we used a high capacity model to model the data distribution in Figure 2, the low density points (Right) would be detected as anomalous. If we used a low capacity model, the cluster (Center) would probably present a higher anomaly score. Our choice of algorithm implicitly imposes a prior on the detected anomalies. Theorem 1 highlights this and makes the need to consider priors explicit.

In a more practicle example, assume we are working with clinical data. In this setting, some low density clusters may indicate diseases (anomalies), while other low density clusters may be caused by uncontrolled factors in the data, such as high performance athletes. At the same time, rare diseases might seem like scattered (low density) points. We want to be able to distinguish between anomalies and 'uninteresting' low probability points.

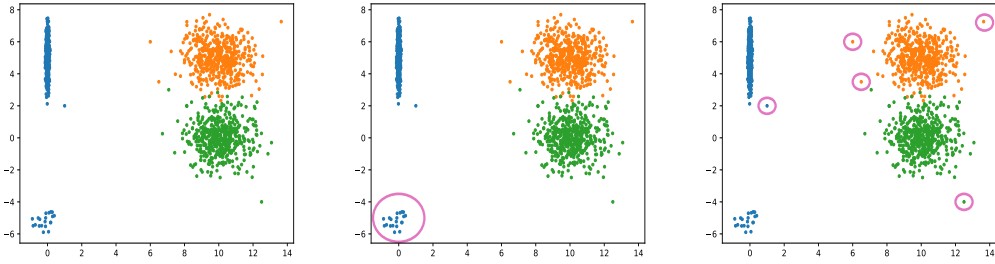

Figure 2: Example of undecidable anomalous data distribution: (Left) Raw data distribution; (Center) Possible Clustered Anomalies; (Right) Possible Low Density Anomalies.

## 3 MODEL

The usual strategy when working with unsupervised anomaly detection problems is training a parameterized model $p_\theta(x)$ to capture the full data distribution $p_{full}(x)$ (e.g. a PCA, or AutoEncoder), and, since $\lambda$ is, by definition, a small constant, assuming $p_{full}(x) \approx p_{normal}(x)$ and assuming points with low probability are anomalous (Zhou & Paffenroth, 2017). An anomaly score $s(x)$ is then defined as $s(x) = \frac{1}{p(x)}$.

There are three main problems with this strategy: (1) if anomalous items are more common than expected, $p_{full}$ might be a poor approximation of $p_{normal}$; (2) if anomalous items are tightly clustered in some way, high capacity models may learn to identify that cluster as a high probability region; (3) if anomalous items are as rare as expected, since we only have access to $p_{full}$, Theorem 1 states we have no information about $p_{anom}$ without further assumptions on its probability distribution.

Most unsupervised anomaly detection systems also already rely on further verification of the results by human experts, due to their uncertain performance. Being mostly used as a ranking system to get high probability instances in the top of a 'list' to be further audited by these experts.

From Theorem 1, we conclude it is impossible to have an universal and reliable unsupervised anomaly detection system, while we know that most such systems already rely on the data being later audited by human experts. These arguments together argue in favor of an active learning strategy for anomaly detection, including the auditor experts in the system's training loop. Thus, anticipating feedback and

benefiting from it to find further anomalous instances, which results in a more robust system. Having an extremely unbalanced dataset in this problem ($\lambda \approx 0$) is also another justification for an active learning setting, which has the potential of requiring exponentially less labeled data than supervised settings (Settles, 2012).

### 3.1 ACTIVE ANOMALY DETECTION

With these motivations, we argue in favor of *active anomaly detection* methods, which despite its many advantages remains an under-explored approach to this problem. Nonetheless, recent work has shown promising results (Veeramachaneni et al., 2016; Das et al., 2016; 2017). In unsupervised anomaly detection, we start with a dataset $\mathcal{D} = \{x | x \sim p_{full}(x)\}$ and want to rank elements in this dataset so that we have the highest possible recall/precision for a certain budget $b$, which is the number of elements selected to be audited by an expert, with no prior information on anomaly labels.

In active anomaly detection, we also start with a completely unlabeled anomaly detection dataset $\mathcal{D} = \{x | x \sim p_{full}(x)\}$, but instead of ranking anomalies and sending them all to be audited at once by our expert, we select them in small parts, waiting for the experts feedback before continuing. We iteratively select the most probable $k \ll b$ elements to be audited[2], wait for the expert to select their label, and continue training our system using this information, as shown in Algorithm 1. This requires the same budget $b$ as an unsupervised anomaly detection system, while having the potential of achieving a much better performance.

---

**Algorithm 1** Active Anomaly Detection

1: **procedure** ACTIVEANOMALYDETECTION($\mathcal{D}$, expert, $b$, $k$)
2:     $i \leftarrow 0$; *labels* $\leftarrow \emptyset$
3:     **while** $i < b$ **do**
4:         *model*.train($\mathcal{D}$, *labels*)
5:         *top_k* $\leftarrow$ *model*.select_top($k$, $\mathcal{D}$, *labels*)
6:         *labels* $\leftarrow$ *labels* $\cup$ *expert*.audit(*top_k*)
7:         $i \leftarrow i + k$

---

With this in mind, we develop the Universal Anomaly Inference (UAI) layer. This layer can be incorporated on top of any deep learning based white box anomaly detection system which provides an anomaly score for ranking anomalies. It takes as input both a latent representation layer ($l(x)$), created by the model, and its output anomaly score ($s(x)$), and passes it through a classifier to find an item's anomaly probability.

$$s_{uai}(x) = p_{anom}(x) = classifier([l(x); s(x)]) \qquad (2)$$

This is motivated by recent works stating learned representations have a simpler statistical structure (Bengio et al., 2013), which makes the task of modeling this manifold and detecting unnatural points much simpler (Lamb et al., 2018). In this work, we model the UAI layer using a simple logistic regression as our classifier, but any architecture could be used here:

$$s_{uai}(x) = p_{anom}(x) = \sigma(W_{act}[l(x); s(x)] + b_{act}) \qquad (3)$$

where $W_{act} \in \mathbb{R}^{1,d+1}$ is a linear transformation, $b_{act} \in \mathbb{R}$ is a bias term and $\sigma(\cdot)$ is the sigmoid function. We learn the values of $W$ and $b$ using back-propagation with a cross entropy loss function, while allowing the gradients to flow through $l$, but not through $s$, since $s$ might be non-differentiable. For the rest of this document, we will refer to the networks with a UAI layer as UaiNets. An example of this architecture is shown in Figure 1.

## 4 EXPERIMENTS

In this section, we test our new UAI layer on top of two distinct architectures: a Denoising AutoEncoder (DAE, with $s_{dae}(x) = ||x - \hat{x}||_2^2$) and a Classifier ($Class$, with $s_{class}(x) =$

---

[2]Although this might seem like a simplistic approach, selecting the top k anomalies is a good strategy in practical settings, since we want to have the most anomalies for any given budget. Besides, because anomaly detection is already a highly imbalanced problem, we might not get anomalous instances even when picking only the top anomalous results, so actively searching for them is usually a good choice. This approach follows recent work in active anomaly detection (Veeramachaneni et al., 2016; Das et al., 2016; 2017)

$cross\_entropy(x_y, \widehat{x_y})$), which use standard multi layer perceptrons. Both architectures are described in details in Appendix A.1. To test our algorithm we start by analyzing its performance on synthetic data created with different properties (Section 4.1). We then present results using UaiNets on real anomaly detection datasets (Section 4.2) and in a semi-supervised setting (Section 4.3).

## 4.1 SYNTHETIC DATA

When designing experiments, we had the objective of showing that our model can work with different definitions of anomaly, while completely unsupervised models will need, by definition, to trade-off accuracy in one setting for accuracy in the other. While this may seem straight forward, these results can show how robust our approach is to the choice of underlying architecture, analyzing how well they do when their underlying architecture has a bad prior for that specific "type" of anomaly. With this in mind, we used the MNIST dataset and defined four sets of experiments:[3]

1. **MNIST$_0$**: For the first set of experiments, we reduced the presence of the 0 digit class to only 10% of its original number of samples, making it only $1/91 \approx 1.1\%$ of the dataset samples. The 0s still present in the dataset had its class randomly changed to $x_y \sim Uniform([1; 9])$ and were defined as anomalies.

2. **MNIST$_{0-2}$**: The second set of experiments follows the same dataset construction, but we reduce the number of instances of numbers 0, 1 and 2, changing the labels of the remaining items in these categories to $x_y \sim Uniform([3; 9])$, and again defining them as anomalous. In this dataset anomalies composed $3/73 \approx 4.1\%$ of the dataset.

3. **MNIST$_{hard}$**: The third set of experiments aims to test a different type of anomaly. In order to create this dataset, we first trained a weak one hidden layer MLP classifier on MNIST and selected all misclassified instances as anomalous, keeping them in the dataset with their original properties ($x_x$ and $x_y$). In this dataset anomalies composed $\approx 3.3\%$ of the dataset.

4. **MNIST$_{pca}$**: In this set of experiments, for each image class ($x_y$), we used a PCA to reduce the dimensionality of MNIST images ($x_x$) to 2 and selected the 5% instances with the largest reconstruction error as anomalies. We kept all 60,000 instances in the dataset with their original properties ($x_x$ and $x_y$) and in this dataset anomalies composed 5% of the dataset.

Results for these experiments are shown in Figure 3 and the main conclusion taken from them is that, even though our algorithm might not get better results than its underlying model for every budget-dataset pair, it is robust to different types of anomalies, which is not the case for the underlying completely unsupervised models. While $Class$ gives really good results in MNIST$_0$ and MNIST$_{0-2}$ datasets, it does not achieve the same performance in MNIST$_{hard}$ and MNIST$_{pca}$, which might indicate it is better at finding clustered anomalies than low density ones. At the same time, $DAE$ has good results for MNIST$_{pca}$ and MNIST$_{hard}$, but bad ones for MNIST$_0$ and MNIST$_{0-2}$, which indicates it is better at finding low density anomalies than clustered ones. Nevertheless, both UaiNets are robust in all four datasets, being able to learn even on datasets which are hard for their underlying models, although they might have a cold start to produce results.

## 4.2 REAL DATA

Here we analyze our model's performance on public benchmarks composed of real anomaly detection datasets. We employ 11 datasets in our analysis: KDDCUP; Thyroid; Arrhythmia; KDDCUP-Rev; Yeast; Abalone; Cardiotocography (CTG); Credit Card; Covtype; Mammography (MMG); Shuttle (Lichman, 2013; Dheeru & Taniskidou, 2017; Pozzolo et al., 2015; Woods et al., 1993). We compare our algorithm against: DAE (Vincent et al., 2008); DAGMM (Zong et al., 2018); LODA-AAD (Das et al., 2016); and Tree-AAD (Das et al., 2017).[4]

---

[3]Implementation details, such as the used architecture and hyper-parameters can be found in Appendix A, as well as further details about the synthetic MNIST datasets. Using MNIST for the generation of synthetic anomaly detection datasets follows recent works (Zhou & Paffenroth, 2017; Zhai et al., 2016). Due to lack of space we only report full results here, but the same plots zoomed in for small budgets ($b \leq 5000$) can be found in Appendix B.1. We also report the same experiments with similar results on Appendix B.3 for the MNIST-Fashion dataset.

[4]Further descriptions of these datasets and baselines can be found in Appendix A.3, as well as descriptions of the used architectures and hyper-parameters. More detailed results, standard deviations and comparison to other baselines are presented in Appendix B.2.

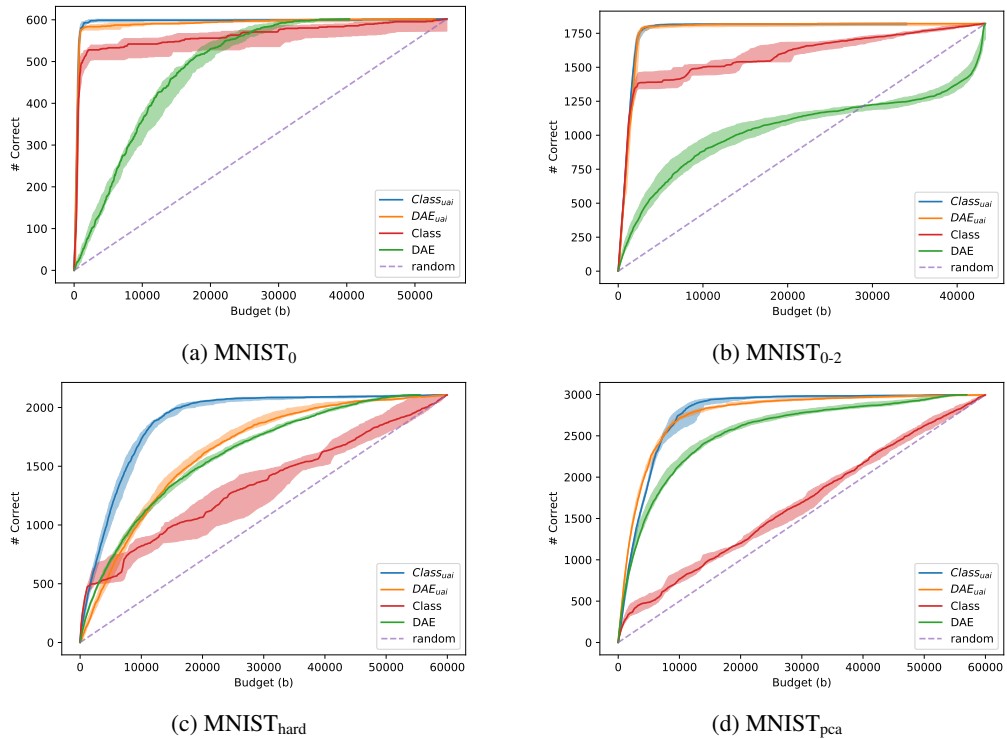

(a) MNIST$_0$

(b) MNIST$_{0-2}$

(c) MNIST$_{hard}$

(d) MNIST$_{pca}$

Figure 3: (Color online) Results for different MNIST experiments. Lines represent median of five runs with different seeds and confidence intervals represent max and min results for each budget $b$.

Table 1 presents results for these real datasets. In these experiments, DAGMM (clean) was trained on a semi-supervised anomaly detection setting, using clean datasets during training, DAGMM (dirty) and DAE were trained in an unsupervised setting, while LODA-AAD, Tree-AAD and $DAE_{uai}$ were trained in an active anomaly detection setting. We can clearly see from these results that DAE produces fairly bad results for all datasets analyzed here, nevertheless, even using a simple architecture as its underlying model, $DAE_{uai}$ produces similar or better results to the best baselines on the 11 datasets, even when the baselines were trained in completely clean training sets. $DAE_{uai}$ also usually presents better results than LODA-AAD and Tree-AAD, which are similarly trained in an active setting.

One possible criticism to our method is that the importance of the proposed approach becomes more relevant the fewer the proportion of anomalous instances, which seems self-defeating. But we see that the largest difference from the active methods to the other algorithms was in Covtype, which has less than 1% anomalies but $286{,}048$ instances. When working with large datasets (>1M instances), even if only 0.1% of the dataset is contaminated there is still the chance to benefit from this feedback to improve performance. The active algorithms are also more robust than the others, DAGMM used different hyperparameters for each experiment, while $DAE_{uai}$ and AAD use the same for all (except for k which was reduced from 10 to 3 for the datasets with less than 100 anomalies).

### 4.3 A MORE PRACTICAL ANOMALY DETECTION SETTING

Another practical scenario where our model could be applied is a mixture of semi-supervised and unsupervised anomaly detection. In this case, we have a dataset which contains anomalies that we want to find and audit. At the same time, new data instances, which may include new types of anomalies not seen before, can be added to the dataset at any time and we would like to detect anomalies in this dataset as well.

$$\mathcal{D}_{train} = X_{train} \sim p_{full}(x) = (1-\lambda)p_{normal}(x) + \lambda p_{anom}(x)$$
$$\mathcal{D}_{test} = X_{test} \sim p_{full}(x) = (1-\lambda_1-\lambda_2)p_{normal}(x) + \lambda_1 p_{anom}(x) + \lambda_2 p_{anom\_new}(x)$$

With this in mind, we ran an experiment training $DAE_{uai}$ and LODA-AAD on KDDCUP-Rev in the same way as in Section 4.2, while evaluating it on its test set for different budgets. This test set

Table 1: Results on Real Datasets showing average F1 scores of five independent runs.

| Train Set | KDDCUP | Arrhythmia | Thyroid | KDDCUP-Rev | Yeast |
|---|---|---|---|---|---|
| # Instances | 494,021 | 3,772 | 452 | 121,597 | 1,191 |
| # Features | 120 | 6 | 274 | 120 | 8 |
| # Anomalies (%) | 97,278 (20%) | 93 (2.5%) | 66 (15%) | 24,319 (20%) | 55 (4.6%) |
| DAGMM (clean) | **0.94** | **0.50** | 0.44 | **0.94** | 0.11 |
| DAGMM (dirty) | 0.43 | 0.46 | 0.46 | 0.31 | 0.02 |
| LODA-AAD | 0.88 | 0.45 | 0.51 | 0.83 | 0.31 |
| Tree-AAD | 0.89 | 0.29 | **0.86** | 0.50 | 0.32 |
| DAE | 0.39 | 0.35 | 0.09 | 0.16 | 0.23 |
| $DAE_{uai}$ | **0.94** | **0.47** | 0.57 | **0.91** | **0.33** |

| Train Set | Abalone | CTG | Credit Card | Covtype | MMG | Shuttle |
|---|---|---|---|---|---|---|
| # Instances | 1,920 | 1,700 | 284,807 | 286,048 | 11,183 | 12,345 |
| # Features | 9 | 22 | 30 | 54 | 6 | 9 |
| # Anomalies (%) | 29 (1.5%) | 45 (2.6%) | 492 (0.17%) | 2,747 (0.9%) | 260 (2.3%) | 867 (7.0%) |
| DAGMM (clean) | 0.16 | 0.27 | 0.34 | 0.18 | 0.07 | 0.48 |
| DAGMM (dirty) | 0.05 | 0.18 | 0.31 | 0.01 | 0.00 | 0.48 |
| LODA-AAD | 0.54 | 0.52 | 0.57 | **0.97** | 0.42 | **0.97** |
| Tree-AAD | 0.53 | **0.69** | **0.76** | 0.94 | 0.59 | 0.92 |
| DAE | 0.08 | 0.13 | 0.36 | 0.15 | 0.27 | 0.17 |
| $DAE_{uai}$ | **0.55** | 0.66 | 0.64 | 0.86 | **0.60** | 0.93 |

contains 20 new types of anomalies (the train set contains 16 types of anomalies and the test set 36). The evaluation was done by selecting the most anomalous instances found by each model on the test set and calculating the recall for both seen and unseen anomalies in that group. Results for this experiment can be seen in Figure 4. In this figure, the right y axis shows the number of anomalies detected in the training set for a certain budget and corresponds to the light blue lines. The left y axis present the recall for the test dataset. We see that DAGMM is not so effective on this test set, while DAE is able to detect well novelty (new classes). We also see that $DAE_{uai}$ is significantly better at detecting known types of anomalies, while it maintains a recall close to the best on new unseen classes, giving better results than LODA-AAD for both seen and unseen classes of anomalies.

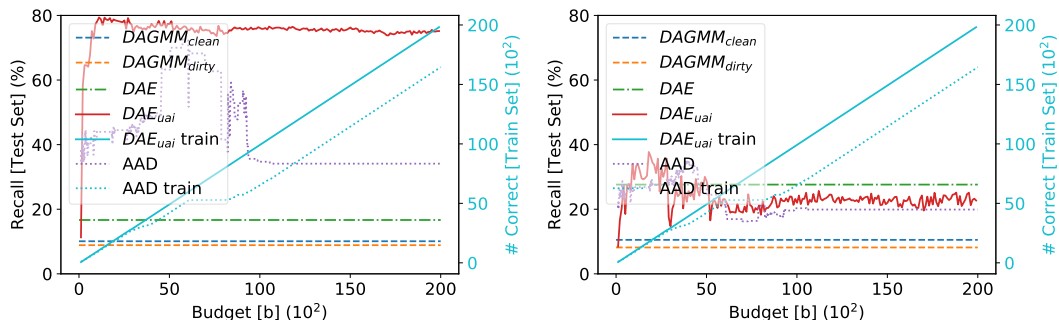

Figure 4: Semi supervised experiment. This test set contains 67,908 instances, out of which 3,817 are previously seen anomalies and 3,498 unseen, totaling 7,315 anomalies. These results show how the Recall@7315 on the test set improve, for both seen and unseen classes of anomalies, as the budget increases during the active training.(Left) Anomalies in training set. (Right) New (unseen) anomalies.

## 5 RELATED WORKS

**Anomaly Detection** This field has been amply studied and good overviews can be found in (Hodge & Austin, 2004; Chandola et al., 2009). Although many algorithms have been recently proposed, classical methods for outlier detection, like LOF Breunig et al. (2000) and OC-SVM (Schölkopf et al., 2001), are still used and produce good results. Recent work on anomaly detection has focused on statistical properties of "normal" data to identify these anomalies, such as Maurus & Plant (2017), which uses Benford's Law to identify anomalies in social networks, and (Siffer et al., 2017), which uses Extreme Value Theory to detect anomalies. Other works focus on specific types of data, (Zheng

et al., 2017) focuses on spatially contextualized data, while (Perozzi et al., 2014; Perozzi & Akoglu, 2016; Li et al., 2017; Liu et al., 2017) focus on graph data. Recently, energy based models (Zhai et al., 2016) and GANs (Schlegl et al., 2017) have been successfully used to detect anomalies, but autoencoders are still more popular in this field. Zhou & Paffenroth (2017) propose a method to train robust autoencoders, drawing inspiration from robust statistics (Huber, 2011) and more specifically robust PCAs, (Yang et al., 2017) focuses on clustering, and trains autoencoders that generate latent representations which are friendly for k-means. The work most similar to ours is DAGMM (Zong et al., 2018), where they train a deep autoencoder and use its latent representations, together with its reconstruction error, as input to a second network, which they use to predict the membership of each data instance to a mixture of gaussian models, training the whole model end-to-end in an semi-supervised manner for novelty detection.

**Active Anomaly Detection** Despite its many advantages, active anomaly detection remains an under-explored approach to this problem, nevertheless, over the years some really interesting work has been developed in this topic. In (Pelleg & Moore, 2005), the authors solve the rare-category detection problem by proposing an active learning strategy to datasets with extremely skewed distributions of class sizes. Abe et al. (2006) reduces outlier detection to classification using artificially generated examples that play the role of potential outliers and then applies a selective sampling mechanism based on active learning to the reduced classification problem. In (Görnitz et al., 2013), the authors proposed a Semi-Supervised Anomaly Detection (SSAD) method based in Support Vector Data Description (SVDD) (Tax & Duin, 2004), which he expanded to a semi-supervised setting, where he accounts for the presence of labels for some anomalous instances, and with an active learning approach to select these instances to label. Veeramachaneni et al. (2016) propose an active approach that combines unsupervised and supervised learning to select items to be labeled by experts, with each approach selecting $\frac{k}{2}$ instances at a time. The most similar prior works to ours in this setting are (Das et al., 2016), which proposed an algorithm that can be employed on top of any ensemble methods based on random projections, and (Das et al., 2017), which expands Isolation Forests to work in an active setting. Our work differs from these prior works mainly in that we prove the necessity of priors for unsupervised anomaly detection, further motivating the Active Anomaly Detection framework, and in our proposed model. UAI layers can be assembled on top of any Deep Learning based anomaly detection architecture, which is the state of the art for unsupervised anomaly detection, to make it work in an active anomaly detection setting. Besides, after each iteration with experts both LODA-AAD and Tree-AAD have a time complexity $O(t)$, where $t$ is the number of already labeled instances, while each iteration of UaiNets runs in constant time $O(1)$ with respect to $t$.

## 6 DISCUSSIONS AND FUTURE WORK

We proposed here a new architecture, Universal Anomaly Inference (UAI), which can be applied on top of any deep learning based anomaly detection architecture. We show that, even on top of very simple architectures, like a DAE, UaiNets can produce similar/better results to state-of-the-art unsupervised/semi-supervised anomaly detection methods. We also give both theoretical and practical arguments motivating *active anomaly detection*, arguing that, in most practical settings, there would be no detriment to using this instead of a fully unsupervised approach.

We further want to make clear that we are not stating our method is better than our semi-supervised baselines (DAGMM, DCN, DSEBM-e). Our contributions are orthogonal to theirs. We propose a new approach to this hard problem which can be built on top of them, this being our main contribution in this work. To the best of our knowledge, this is the first work which applies deep learning to active anomaly detection. We use the strongest points of these deep learning algorithms (their learned representations and anomaly scores) to build an active algorithm, presenting an end-to-end architecture which learns representations by leveraging both the full dataset and the already labeled instances.

Important future directions for this work are using the UAI layers confidence in its output to dynamically choose between either directly using its scores, or using the underlying unsupervised model's anomaly score to choose which instances to audit next. Another future direction would be testing new architectures for UAI layers, in this work we restricted all our analysis to simple logistic regression. A third important future work would be analyzing the robustness of UaiNets to mistakes being made by the labeling experts. Finally, making this model more interpretable, so that auditors could focus on a few "important" features when labeling anomalous instances, could increase labeling speed and make their work easier.

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

# A    Experiments Descriptions

In this section we give detailed descriptions of the experiments. Section A.1 presents the used model architectures for both $DAE$ and $Class$ models, as well as $DAE_{uai}$ and $Class_{uai}$. Section A.2 presents details on the synthetic MNIST datasets and on the hyper-parameters used for the experiments. Finally, Section A.3 contains detailed descriptions on the used datasets, baselines and experimental settings for the experiments on real anomaly detection datasets.

## A.1    Model Architectures

To show our algorithm can be assembled on top of any deep learning model, we tested it using two simple but very different anomaly detection models. The first model we test it on top of is a normal Denoising AutoEncoder (DAE). A DAE is a neural network mainly composed by an encoder, which transforms the input into a latent space, and a decoder, which reconstructs the input using this latent representation, typically having a loss function that minimizes the reconstruction error $L_2$ norm:

$$l = f_{enc}(x + \epsilon) \quad \epsilon \sim \mathcal{N}(0, \varphi)$$
$$\hat{x} = f_{dec}(l) \tag{4}$$
$$\mathcal{L} = ||x - \hat{x}||_2^2$$

where both $f_{enc}$ and $f_{dec}$ are usually feed forward networks with the same number of layers, $l \in \mathbb{R}^d$ is a $d$-dimensional latent representation and $\epsilon$ is a zero mean noise, sampled from a Gaussian distribution with a $\varphi$ standard deviation. When used in anomaly detection, the reconstruction error is usually used as an approximation of the inverse of an item's probability, and as its anomaly score:

$$s_{dae}(x) = \frac{1}{p(x)} = ||x - \hat{x}||_2^2 \tag{5}$$

We then create a $DAE_{uai}$ network by assembling the proposed UAI layer on top of the DAE:

$$l_{dae} = l = f_{enc}(x + \epsilon)$$
$$s_{dae-uai}(x) = uai([l_{dae}; s_{dae}]) \tag{6}$$

where $uai(\cdot)$ is the classifier chosen for the UAI layer. This architecture can be seen in Figure 1. Another typical approach to unsupervised anomaly detection is, when given a dataset with labeled data $X = (x_x, x_y)$, training a classifier ($Class$) to predict $x_y$ from $x_x$[5] and using the cross-entropy of an item as an approximation to the inverse of its probability distribution:

$$\widehat{x_y} = f_{class}(x)$$
$$\mathcal{L} = cross\_entropy(x_y, \widehat{x_y}) \tag{7}$$
$$s_{class}(x) = \frac{1}{p(x)} = cross\_entropy(x_y, \widehat{x_y})$$

where $f_{class}(\cdot)$ is typically a feed forward neural network with $p$ layers, from which we can use its last hidden layer ($h_{p-1}$) as the data's latent representation to be used in the $Class_{uai}$.

$$l_{class} = h_{p-1}$$
$$s_{class-uai}(x) = uai([l_{class}; s_{class}]) \tag{8}$$

This architecture can be seen in Figure 5. For all experiments in this work, unless otherwise stated, the DAE's encoder and decoder had independent weights and we used both the $DAE$ and $Class$ models with 3 hidden layers and hidden sizes $[256, 64, 8]$. This means the latent representations provided to the UAI layers are $l \in \mathbb{R}^8$. We implemented all experiments using TensorFlow (Abadi et al., 2016), and used a learning rate of $0.01$, batch size of $256$ and the RMSprop optimizer with the default hyper-parameters. For the active learning models, we pre-train the DAE/Class model for $5000$ optimization steps, select $k = 10$ items to be labeled at a time, and further train for $100$ iterations after each labeling call. To deal with the cold start problem, for the first 10 calls of select_top, we use the base anomaly score ($s$) of the $DAE/Class$ model to make this selection, using the UAI one for all later labeling decisions.

---

[5]Note that, even though in this problem we have class labels ($x_y$), we have no anomaly labels of objects ($y$), so this is still an unsupervised anomaly detection problem.

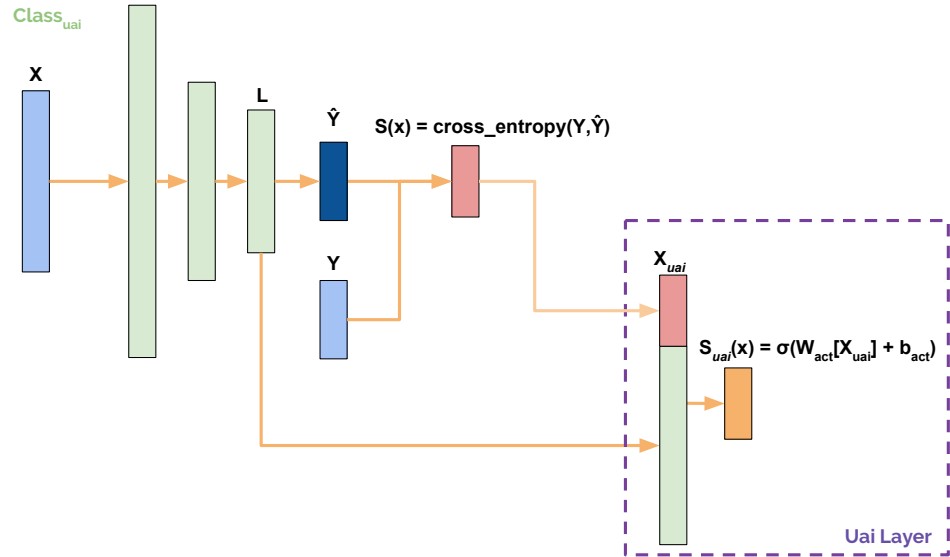

Figure 5: $Class_{uai}$ architecture.

Table 2: MNIST Anomaly Datasets Statistics

|  | # Dimensions | # Classes | # Instances | # Anomalies | Anomaly Ratio |
|---|---|---|---|---|---|
| $\text{MNIST}_0$ | 784 | 9 | 54,679 | 602 | 1.1% |
| $\text{MNIST}_{0\text{-}2}$ | 784 | 7 | 43,199 | 1,822 | 4.2% |
| $\text{MNIST}_{\text{hard}}$ | 784 | 10 | 60,000 | 2,108 | 3.5% |
| $\text{MNIST}_{\text{pca}}$ | 784 | 10 | 60,000 | 2,996 | 5% |

## A.2 SYNTHETIC DATA

Detailed statistics on the synthetic MNIST datasets can be seen in Table 2. $\text{MNIST}_0$ and $\text{MNIST}_{0\text{-}2}$ were mainly generated with the purpose of simulating the situation in Figure 2 (Center), where anomalies were present in sparse clusters. At the same time, $\text{MNIST}_{\text{hard}}$ and $\text{MNIST}_{\text{pca}}$ were designed to present similar characteristics to the situation in Figure 2 (Right), where anomalous instances are in sparse regions of the data space.

## A.3 REAL DATA

For these experiments, most datasets were used as suggested in (Dheeru & Taniskidou, 2017), but we processed the KDDCUP, Thyroid, Arrhythmia and KDDCUP-Rev datasets in the same manner as (Zong et al., 2018) to be able to better compare with their results:

- **KDDCUP (Lichman, 2013)**: The KDDCUP99 10 percent dataset from the UCI repository. Since it contains only 20% of instances labeled as "normal" and the rest as "attacks", "normal" instances are used as anomalies, since they are in a minority group. This dataset contains 34 continuous features and 7 categorical ones. We transform these 7 categorical features into their one hot representations, and obtain a dataset with 120 features.

- **Thyroid (Lichman, 2013)**: A dataset containing data from patients which can be divided in three classes: normal (not hypothyroid), hyperfunction and subnormal functioning. In this dataset, we treat the hyperfunction class as an anomaly, with the other two being treated as normal. It can be obtained from the ODDS repository.[6]

- **Arrhythmia (Lichman, 2013)**: This dataset was designed to create classification algorithms to distinguish between the presence and absence of cardiac arrhythmia. In it, we use the

---

[6]http://odds.cs.stonybrook.edu

      smallest classes (3, 4, 5, 7, 8, 9, 14, and 15) as anomalies and the others are treated as normal. This dataset can also be obtained from the ODDS repository.

- **KDDCUP-Rev (Lichman, 2013)**: Since "normal" instances are a minority in the KDDCUP dataset, we keep all "normal" instances and randomly draw "attack" instances so that they compose 20% of the dataset.

We compare our algorithm against:

- **DAE** (Vincent et al., 2008): Denoising Autoencoders are autoencoder architectures which are trained to reconstruct instances from noisy inputs.

- **DAGMM** (Zong et al., 2018): Deep Autoencoding Gaussian Mixture Model is a state-of-the-art model for semi-supervised anomaly detection which simultaneously learns a latent representation, using deep autoencoders, and uses both this latent representation and the autoencoder's reconstruction error to learn a Gaussian Mixture Model for the data distribution.

- **LODA-AAD** (Das et al., 2016): Lightweight on-line detector of anomalies (LODA) Active Anomaly Discovery (AAD) is a work which uses the active anomaly detection framework on top of LODA (Pevnỳ, 2016), which is a method based on ensembles of weak anomaly detection models.

- **Tree-AAD** (Das et al., 2017): This work learns weights for each node in an Isolation Forest anomaly detection model, by incorporating knowledge gained through active anomaly detection.

Since there is no validation/test set in unsupervised anomaly detection, we cannot tune hyperparameters on a validation set. Because of this, to make the DAE baselines more competitive, we got the results for several different hyper-parameter configurations and present only the best among them. This is not a realistic approach, but we only do it to our baselines, while for our proposed algorithm we keep hyper-parameters fixed for all experiments. We even keep our hidden sizes fixed to $[256, 64, 8]$ on thyroid, which only contains 6 features per instance, since our objective here is not getting the best possible results, but showing the robustness of our approach. The only hyper-parameter change we make in UAI networks is that, since there are fewer anomalies in some datasets, we set our active learning approach to choose $k = 3$ instances at a time, instead of 10, for datasets with less than 100 anomalies.

Results for DAGMM are from our implementation of this model and follow the same procedures, architectures and hyper-parameters as described in (Zong et al., 2018), being trained in a semi-supervised setting. The results for LODA-AAD and Tree-AAD were run using the code made available by the authors and with the same steps as $DAE_{uai}$.[7] For all experiments, results for LODA-AAD, Tree-AAD, DAE and $DAE_{uai}$ used the number of anomalies in the dataset as the budget $b$.

## B  DETAILED RESULTS

In this section, we present more detailed results for both the synthetic (Section B.1) and real (Section B.2) anomaly detection datasets, which couldn't fit on the main paper due to lack of space. We also present results for synthetic anomaly detection experiments on Fashion-MNIST (Section B.3).

### B.1  DETAILED RESULTS ON MNIST

We present here detailed results for small budgets ($b \leq 5000$) on the MNIST experiments, with graphs zoomed in for these budget values. Analyzing Figure 6 we see that for some of these datasets UaiNets present a cold start, producing worse results for small budgets. Nonetheless, after this cold start, they produce better results in all MNIST experiments. An interesting future work would be to measure the confidence in the UaiNet's prediction to dynamically choose between using its anomaly score or the underlying network's one, which could solve/reduce this cold start problem.

---

[7]https://github.com/shubhomoydas/ad_examples

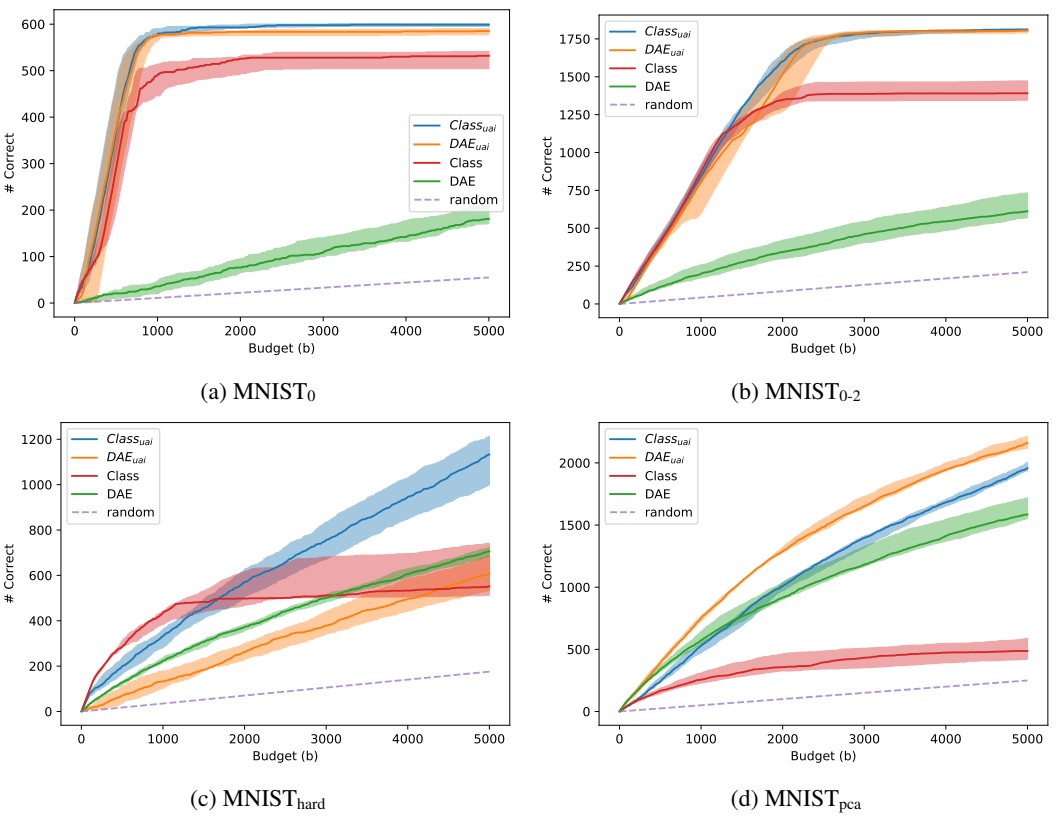

(a) MNIST$_0$          (b) MNIST$_{0-2}$

(c) MNIST$_{hard}$          (d) MNIST$_{pca}$

Figure 6: (Color online) Results for MNIST experiments zoomed in for $b \leq 5000$ on the x-axis. Lines represent median of five runs with different seeds and confidence intervals represent max and min results for each budget $b$.

## B.2 DETAILED RESULTS ON REAL DATA

Table 3 presents a detailed comparison for experiments ran on KDDCUP, Thyroid, Arrhythmia and KDDCUP-Rev datasets with other baselines, also showing precision, recall and their standard deviations. In this table we also compare our results to:

- **OC-SVM** (Chen et al., 2001): One-class support vector machines are a popular kernel based anomaly detection method. In this work, we employ it with a Radial Basis Function (RBF) kernel.

- **DCN** (Yang et al., 2017): Deep Clustering Network is a state-of-the-art clustering algorithm. Its architecture is designed to learn a latent representation using deep autoencoders which is easily separable when using k-means.

- **PAE** (Vincent et al., 2008): Denoising AutoEncoders pretrained as suggested in (Vincent et al., 2010).

- **DSEBM-e** (Zhai et al., 2016): Deep Structured Energy Based Models are anomaly detection systems based on energy based models (LeCun et al., 2006), which are a powerful tool for density estimation. We compare here against DSEBM-e, which uses a data instance's energy as the criterion to detect anomalies.

- **DSEBM-r** (Zhai et al., 2016): Deep Structured Energy Based Model with the same architecture and training procedures as DSEBM-e, but using an instance's reconstruction error as the criterion for anomaly detection.

The results presented here are averages of five runs, with standard deviations in parenthesis. In this table, results for OC-SVM, PAE, DSEBM-r, DSEBM-e, DCN and DAGMM were taken from (Zong

Table 3: Detailed results on real datasets showing mean and standard deviations of five runs.

| Dataset | Method | Anomalies in Train Set | Precision | Recall | F1 |
|---|---|---|---|---|---|
| KDDCUP | OC-SVM | 0% | 0.7457 | 0.8523 | 0.7954 |
| | OC-SVM | 5% | 0.1155 | 0.3369 | 0.1720 |
| | PAE | 0% | 0.7276 | 0.7397 | 0.7336 |
| | DSEBM-r | 0% | 0.1972 | 0.2001 | 0.1987 |
| | DSEBM-e | 0% | 0.7369 | 0.7477 | 0.7423 |
| | DSEBM-e | 5% | 0.5345 | 0.5375 | 0.5360 |
| | DCN | 0% | 0.7696 | 0.7829 | 0.7762 |
| | DCN | 5% | 0.6763 | 0.6893 | 0.6827 |
| | DAGMM | 0% | 0.9297 | **0.9442** | 0.9369 |
| | DAGMM | 5% | 0.8504 | 0.8643 | 0.8573 |
| | DAGMM* | 0% | 0.9290 (0.0344) | 0.9435 (0.0349) | 0.9362 (0.0346) |
| | DAGMM* | 5% | 0.8827 (0.0682) | 0.8965 (0.0693) | 0.8896 (0.0688) |
| | DAGMM* | 20% | 0.4238 (0.0187) | 0.4304 (0.0190) | 0.4271 (0.0188) |
| | LODA-AAD | 20% | 0.8756 (0.1255) | 0.8756 (0.1255) | 0.8756 (0.1255) |
| | Tree-AAD | 20% | 0.8940 (0.0261) | 0.8940 (0.0261) | 0.8940 (0.0261) |
| | DAE | 20% | 0.3905 (0.2581) | 0.3905 (0.2581) | 0.3905 (0.2581) |
| | $DAE_{uai}$ | 20% | **0.9401** (0.0191) | **0.9401** (0.0191) | **0.9401** (0.0191) |
| Thyroid | OC-SVM | 0% | 0.3639 | 0.4239 | 0.3887 |
| | PAE | 0% | 0.1894 | 0.2062 | 0.1971 |
| | DSEBM-r | 0% | 0.0404 | 0.0403 | 0.0403 |
| | DSEBM-e | 0% | 0.1319 | 0.1319 | 0.1319 |
| | DCN | 0% | 0.3319 | 0.3196 | 0.3251 |
| | DAGMM | 0% | 0.4766 | 0.4834 | 0.4782 |
| | DAGMM* | 0% | 0.4375 (0.1926) | 0.4468 (0.1967) | 0.4421 (0.1947) |
| | DAGMM* | 0.5% | 0.2875 (0.1505) | 0.2936 (0.1537) | 0.2905 (0.1521) |
| | DAGMM* | 2.5% | 0.4542 (0.2995) | 0.4638 (0.3059) | 0.4590 (0.3027) |
| | LODA-AAD | 2.5% | 0.5097 (0.0712) | 0.5097 (0.0712) | 0.5097 (0.0712) |
| | Tree-AAD | 2.5% | **0.8586** (0.0087) | **0.8586** (0.0087) | **0.8586** (0.0087) |
| | DAE | 2.5% | 0.0860 (0.0725) | 0.0860 (0.0725) | 0.0860 (0.0725) |
| | $DAE_{uai}$ | 2.5% | 0.5742 (0.0582) | 0.5742 (0.0582) | 0.5742 (0.0582) |
| Arrhythmia | OC-SVM | 0% | **0.5397** | 0.4082 | 0.4581 |
| | PAE | 0% | 0.4393 | 0.4437 | 0.4403 |
| | DSEBM-r | 0% | 0.1515 | 0.1513 | 0.1510 |
| | DSEBM-e | 0% | 0.4667 | 0.4565 | 0.4601 |
| | DCN | 0% | 0.3758 | 0.3907 | 0.3815 |
| | GADMM | 0% | 0.4909 | **0.5078** | **0.4983** |
| | GADMM* | 0% | 0.4902 (0.0514) | 0.5051 (0.0530) | 0.4975 (0.0522) |
| | GADMM* | 3% | 0.4530 (0.0573) | 0.4666 (0.0591) | 0.4597 (0.0582) |
| | GADMM* | 15% | 0.4500 (0.0597) | 0.4636 (0.0615) | 0.4567 (0.0606) |
| | LODA-AAD | 15% | 0.4485 (0.0136) | 0.4485 (0.0136) | 0.4485 (0.0136) |
| | Tree-AAD | 15% | 0.2882 (0.0257) | 0.2882 (0.0257) | 0.2882 (0.0257) |
| | DAE | 15% | 0.3485 (0.0392) | 0.3485 (0.0392) | 0.3485 (0.0392) |
| | $DAE_{uai}$ | 15% | **0.4727** (0.0225) | **0.4727** (0.0225) | **0.4727** (0.0225) |
| KDDCUP-Rev | OC-SVM | 0% | 0.7148 | **0.9940** | 0.8316 |
| | PAE | 0% | 0.7835 | 0.7817 | 0.7826 |
| | DSEBM-r | 0% | 0.2036 | 0.2036 | 0.2036 |
| | DSEBM-e | 0% | 0.2212 | 0.2213 | 0.2213 |
| | DCN | 0% | 0.2875 | 0.2895 | 0.2885 |
| | GADMM | 0% | 0.9370 | 0.9390 | 0.9380 |
| | GADMM* | 0% | **0.9391** (0.1553) | 0.9391 (0.1553) | **0.9391** (0.1553) |
| | GADMM* | 5% | 0.3184 (0.1358) | 0.3559 (0.2096) | 0.3341 (0.1658) |
| | GADMM* | 20% | 0.3051 (0.1059) | 0.3053 (0.1060) | 0.3052 (0.1059) |
| | LODA-AAD | 20% | 0.8339 (0.1081) | 0.8339 (0.1081) | 0.8339 (0.1081) |
| | Tree-AAD | 20% | 0.5032 (0.2984) | 0.5032 (0.2984) | 0.5032 (0.2984) |
| | DAE | 20% | 0.1626 (0.0609) | 0.1626 (0.0609) | 0.1626 (0.0609) |
| | $DAE_{uai}$ | 20% | **0.9117** (0.0160) | **0.9125** (0.0170) | **0.9121** (0.0165) |

et al., 2018), while DAGMM* are results from our implementation of DAGMM. Unfortunately, we were not able to reproduce their results in the Thyroid dataset, getting a high variance in the results. LODA-AAD does not scale well to large datasets, so to run it on KDDCUP and KDDCUP-Rev we needed to limit its memory about the anomalies it had already learned, forgetting the oldest ones. This reduced its runtime complexity from $O(b^2)$ to $O(b)$ in our tests, where $b$ is the budget limit for the anomaly detection task. We did the same (limit memory) for Tree-AAD on KDDCUP.

On this table we can see that $DAE_{uai}$ produces better results than LODA-AAD on all analyzed datasets and than Tree-AAD on three out of four. Our proposed method also, besides presenting results comparable to state-of-the-art DAGMM trained on a clean dataset, is much more stable, having a lower standard deviation than the baselines in almost all datasets.

### B.3 Experiments on Fashion-MNIST

In this Section, we present results for experiments on synthetic anomaly detection datasets based on Fashion-MNIST (Xiao et al., 2017). To create these datasets we follow the same procedures as done for MNIST in Section 4.1, generating four datasets: Fashion-MNIST$_0$; Fashion-MNIST$_{0-2}$; Fashion-MNIST$_{\text{hard}}$; Fashion-MNIST$_{\text{pca}}$. Detailed statistics of these datasets can be seen in Table 4.

Table 4: Fashion-MNIST Anomaly Datasets Statistics

|  | # Dimensions | # Classes | # Instances | # Anomalies | Anomaly Ratio |
|---|---|---|---|---|---|
| Fashion-MNIST$_0$ | 784 | 9 | 54,610 | 610 | 1.1% |
| Fashion-MNIST$_{0-2}$ | 784 | 7 | 43,765 | 1,765 | 4.0% |
| Fashion-MNIST$_{\text{hard}}$ | 784 | 10 | 60,000 | 9,656 | 16.1% |
| Fashion-MNIST$_{\text{pca}}$ | 784 | 10 | 60,000 | 3,000 | 5.0% |

We run experiments on these datasets following the exact same procedures as in Section 4.1. Figure 7 shows the results for Fashion-MNIST$_0$ and Fashion-MNIST$_{0-2}$, while Figure 8 show the results for Fashion-MNIST$_{\text{hard}}$ and Fashion-MNIST$_{\text{pca}}$. These figures show similar trends to the ones for MNIST, although algorithms find anomalies in these datasets harder to identify. In one run of Fashion-MNIST$_0$, $DAE_{uai}$ needed several examples to start learning and for Fashion-MNIST$_{\text{hard}}$, $Class_{uai}$ takes a long time to start producing better results than $Class$. Nevertheless, UaiNets are still much more robust than the underlying networks to different types of anomalies, producing good results in all four datasets, even when its underlying network gives weak results on that dataset.

## C   Further Analysis

In this section we further study UaiNets, analyzing the evolution of hidden representations and anomaly scores through training (Section C.1), and the dependence of results on the number of audited anomalies (Section C.2).

### C.1   Learned Representations and Anomaly Scores

In this section, we show visualizations of the learned representations ($l_{dae/class}$) and anomaly scores ($s_{dae/class}$) of UaiNets' underlying networks, presenting their evolution as more labels are fed into the network through the active learning process. With this purpose, we retrain UaiNets on both MNIST$_{0-2}$ and MNIST$_{\text{hard}}$, with a hidden size of $[256, 64, 1]$, so that its latent representation is one dimensional ($l(x) \in R^1$), and plot these representations vs the anomaly scores ($s$) of the base network (either $DAE$ or $Class$) for different budgets ($b$).

Figure 9 shows the evolution of $DAE_{uai}$'s underlying $l_{dae}(x)$ and $s_{dae}(x)$. In it, we can see that initially (Figures 9 (a, d)) anomalies and normal data instances are not separable in this space. Nevertheless, with only a few labeled instances ($b = 250$) the space becomes much easier to separate, while for $b = 2000$ the space is almost perfectly linearly separable.[8]

Figure 10 shows the same evolution for $Class_{uai}$'s underlying $l_{class}(x)$ and $s_{class}(x)$. In it, we can also see the same patterns, as initially anomalies and normal data instances are not separable, but with a few labeled instances anomalies become much more identifiable.

The main conclusion taken from these visualizations is how the gradient flow through $l$ is important, since it helps the network better separate data in these spaces, allowing good performance even when the underlying networks are not good at identifying a specific type of anomaly.

---

[8]Gifs showing this choice evolution will be made available with the final publication.

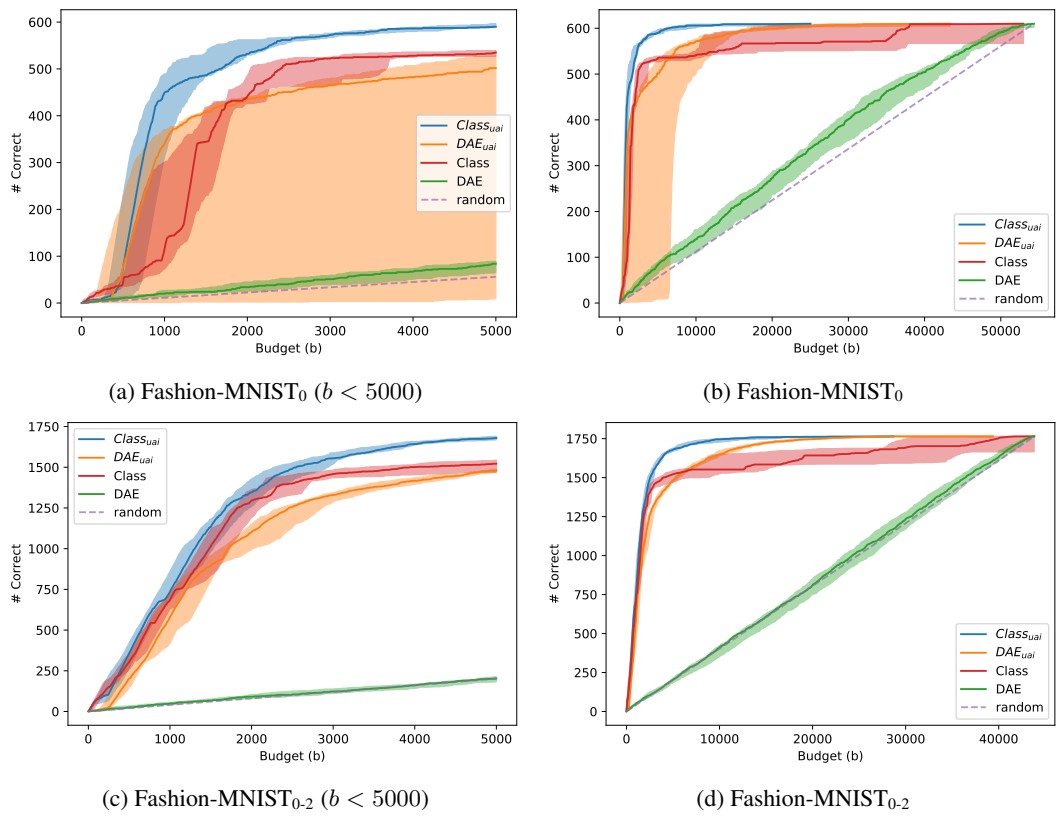

Figure 7: (Color online) Results for Fashion-MNIST$_0$ and Fashion-MNIST$_{0\text{-}2}$ with different zooms on x-axis. Lines represent median of five runs with different seeds and confidence intervals represent max and min results for each budget $b$.

## C.2 ANOMALY CHOICES EVOLUTION THROUGH TRAINING

This experiments aim at showing how the networks choice quality evolves with the access to more labels. Here, we present the choices $DAE_{uai}$ network would make having access to a fixed number of expert labels. With this in mind, we train the networks in the same way as in Section 4.2, but stop after reaching a specific budget ($b$), showing the choices made up to that point, and after that with no further training.

Figure 11 shows the evolution of $DAE_{uai}$ anomaly choices as it is fed more expert knowledge. We can see that with only a few labels it already fairs a lot better than its underlying network. In KDDCUP with only 3,000 labeled instances, which is less than $1\%$ of the dataset, it can correctly find 80,000 anomalies with a high precision, while the $DAE$ with no expert knowledge does a lot worse. On Thyroid and KDDCUP-Rev, with $\approx 10\%$ of the dataset labeled ($b = 531$ and $b = 4000$, respectively) it finds all or almost all anomalies in the dataset correctly. The Arrhythmia dataset is a lot smaller and with few anomalies, so $DAE_{uai}$ improves on $DAE$ in a smaller scale here, but it still does fairly better than the underlying network.[9]

---

[9]Gifs showing this choice evolution will be made available with the final publication.

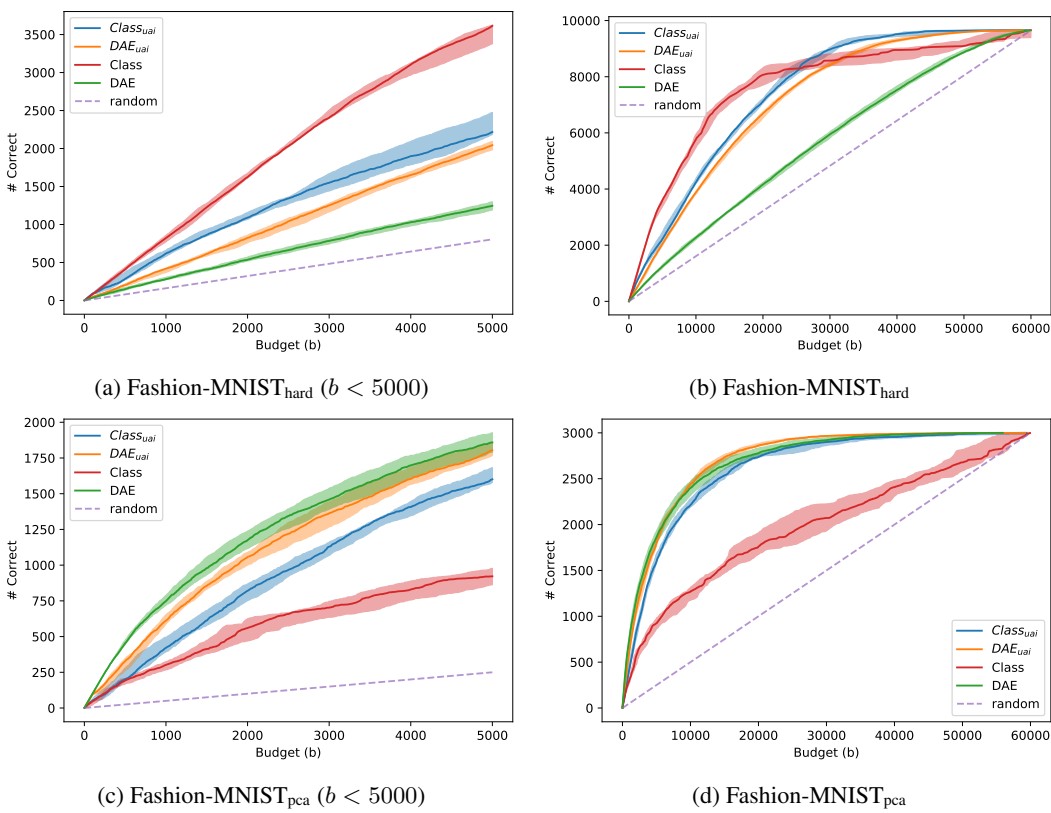

Figure 8: (Color online) Results for Fashion-MNIST$_{\text{hard}}$ and Fashion-MNIST$_{\text{pca}}$ with different zooms on x-axis. Lines represent median of five runs with different seeds and confidence intervals represent max and min results for each budget $b$.

# D    PROOFS

## D.1    LEMMA 1. MIXTURE PROBABILITY LEMMA

**Lemma 1.** ***Mixture probability lemma.*** *Consider two independent arbitrary probability distributions $p_1$ and $p_2$. Given only a third distribution $p_+ = \overline{p}$ composed of the weighted average of the two:*

$$p_+ = (1 - \lambda) \cdot p_1 + \lambda \cdot p_2, \quad 0 \leq \lambda \leq 1$$

*and considering $P_i$ as the residual probability distribution hyperplanes:*

$$
\begin{aligned}
P_1 &= \left\{ p_r = \tfrac{\overline{p} - \lambda \cdot p}{1 - \lambda}, \forall p \in P \mid \lambda \in [0;1], \lambda \cdot p \leq \overline{p} \right\} \\
&= \left\{ p_r, \forall p_r \in P \mid \lambda \in [0;1], (1 - \lambda) \cdot p_r \leq \overline{p} \right\} \\
P_2 &= \left\{ p_r = \tfrac{\overline{p} - (1 - \lambda) \cdot p}{\lambda}, \forall p \in P \mid \lambda \in [0;1], (1 - \lambda) \cdot p \leq \overline{p} \right\} \\
&= \left\{ p_r, \forall p_r \in P \mid \lambda \in [0;1], \lambda \cdot p_r \leq \overline{p} \right\}
\end{aligned}
$$

*Without further assumptions on $p_2$ (without a prior on its probability distribution), we only know that $p(p_1 | p_+ = \overline{p}) = p(p_1 | p_1 \in P_1)$ and $p(p_2 | p_+ = \overline{p}_\alpha) = p(p_1 = \overline{p}_\alpha - \lambda_\alpha \cdot p_2 | p_2 \in P_2)$.*

*Proof.* Given $p_+ = \overline{p}$ we know that:

$$p_1 + \lambda_\alpha \cdot p_2 = \overline{p}_\alpha$$

with $\lambda_\alpha = \frac{\lambda}{1 - \lambda}$ and $\overline{p}_\alpha = \frac{\overline{p}}{1 - \lambda}$. Assuming the distribution of $p_2$ is independent of $p_1$, and with no further assumptions on it, $p_2$ is random and uniform on the set of all possible probability distributions,

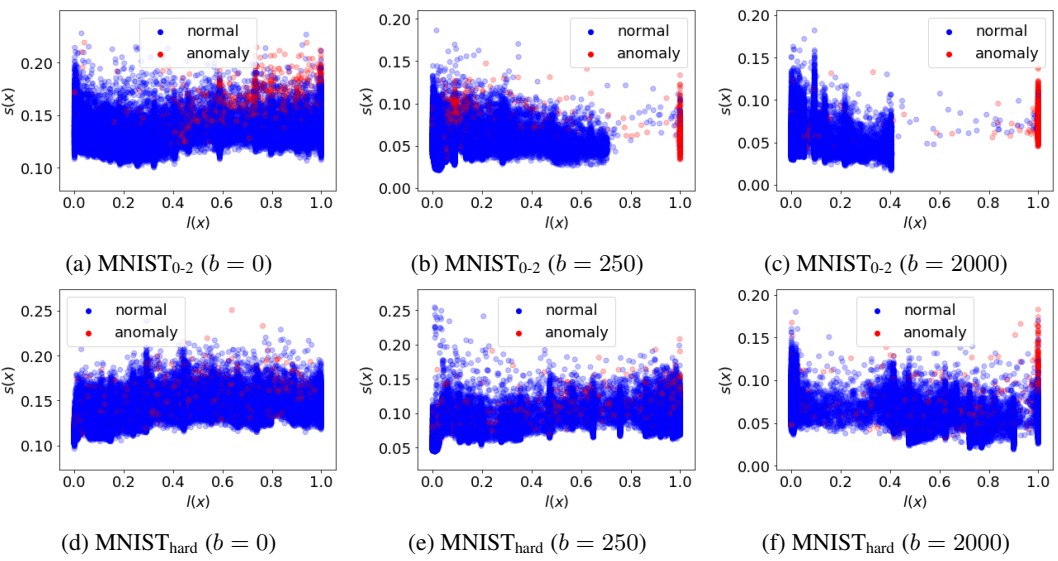

Figure 9: (Color online) Underlying latent representations ($l_{dae}$) vs anomaly score ($s_{dae}$) for $DAE_{uai}$ network as training progresses on MNIST$_{0\text{-}2}$ and MNIST$_{hard}$.

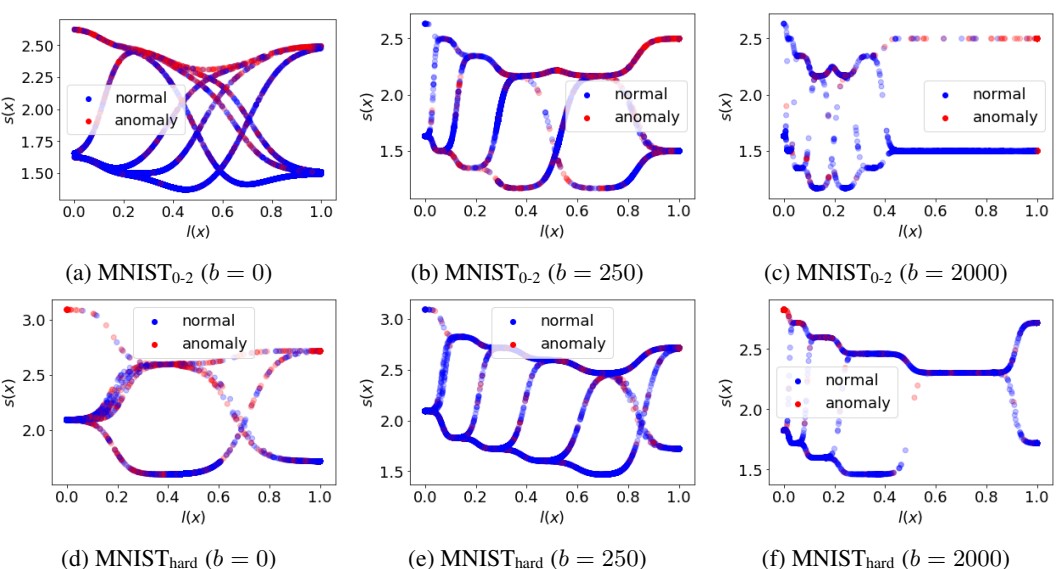

Figure 10: (Color online) Underlying latent representations ($l_{class}$) vs anomaly score ($s_{class}$) for $Class_{uai}$ network as training progresses on MNIST$_{0\text{-}2}$ and MNIST$_{hard}$.

so its probability distribution is:

$$p_2 \sim Uniform(P)$$

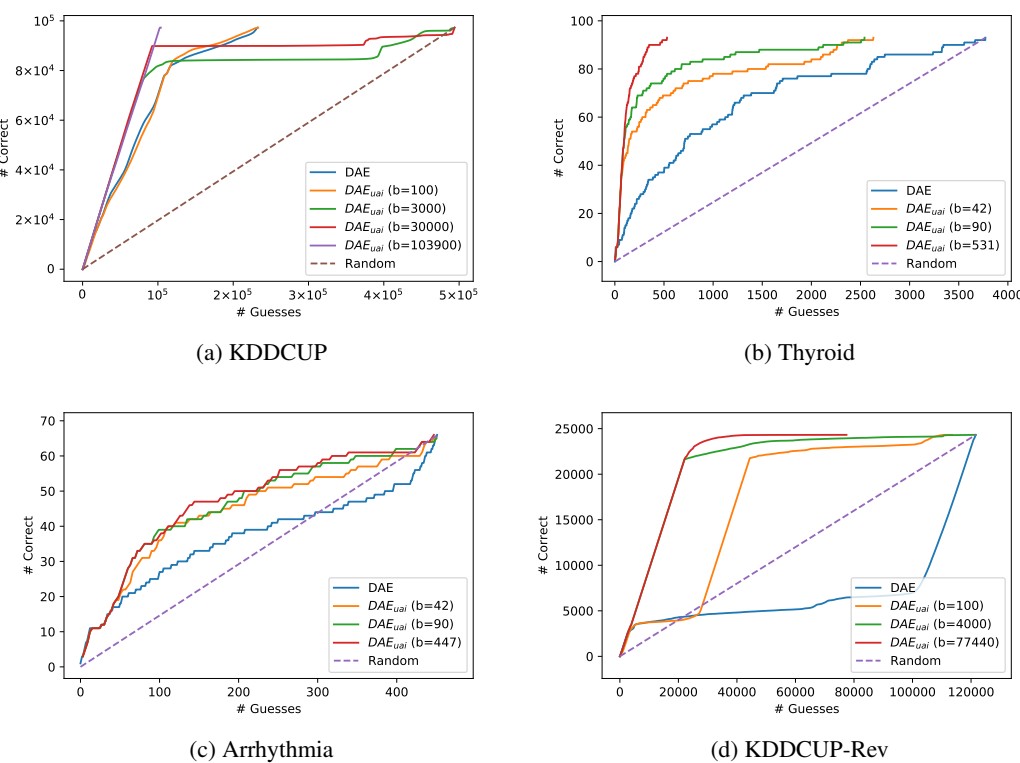

(a) KDDCUP

(b) Thyroid

(c) Arrhythmia

(d) KDDCUP-Rev

Figure 11: (Color online) Results for the real anomaly detection datasets when the UaiNets are only fed expert information until a budget ($b$) limit. Lines stop in the x-axis when all anomalies have been discovered.

where $P$ is the hyperspace containing all probability distributions, with an hyper-volume $m$. Now we can try to find $p(p_1|p_+ = \overline{p})$:

$$
\begin{aligned}
p(p_1|p_+ = \overline{p}) &= p\left(p_+ = \overline{p}|p_1\right) \cdot \frac{p(p_1)}{p(p_+ = \overline{p})} \\
&\overset{(1)}{=} p\left(p_+ = \overline{p}|p_1\right) \cdot \frac{p(p_1)}{p(p_+ = \overline{p})} \qquad\qquad\qquad , p_1 \in P_1 \\
&\overset{(2)}{=} p\left(p_2 = \frac{\overline{p}_\alpha - p_1}{\lambda_\alpha}\Big|p_1\right) \cdot \frac{p(p_1)}{\int_x p\left(p_1 = x, p_2 = \frac{\overline{p}_\alpha - x}{\lambda_\alpha}\right)dx} \\
&\overset{(3)}{=} p\left(p_2 = \frac{\overline{p}_\alpha - p_1}{\lambda_\alpha}\right) \cdot \frac{p(p_1)}{\int_x p\left(p_1 = x, p_2 = \frac{\overline{p}_\alpha - x}{\lambda_\alpha}\right)dx} \\
&\overset{(4)}{=} \frac{1}{m} \cdot \frac{p(p_1)}{\int_x p\left(p_1 = x|p_2 = \frac{\overline{p}_\alpha - x}{\lambda_\alpha}\right) \cdot p\left(p_2 = \frac{\overline{p}_\alpha - x}{\lambda_\alpha}\right)dx} \\
&\overset{(5)}{=} \frac{1}{m} \cdot \frac{p(p_1)}{\int_{x \in P_1} p(p_1 = x) \cdot \frac{1}{m}dx + \int_{x \notin P_1} 0 dx} \\
&= \frac{p(p_1)}{\int_{x \in P_1} p(p_1 = x)dx} \\
p(p_1|p_+ = \overline{p}_\alpha) &= p(p_1|p_1 \in P_1)
\end{aligned}
$$

The equality in (1) comes from the definition of the space $P_1$, which is the space of all possible values of $p_1$ that could result in $p_+ = \overline{p}$, so if $p_1 \notin P_1$, then $p\left(p_+ = \overline{p}|p_1\right) = 0$. Equality (2) is a simple

variable substitution where $p\left(p_+ = \overline{p}\right) = p\left(p_1 = x, p_2 = \frac{\overline{p}_\alpha - x}{\lambda_\alpha}\right)$. (3) comes from the assumption that $p_2$ and $p_1$ are independent. Equality (4) results from $p_2 \sim Uniform(P)$ and $P$ having a volume $m$. Finally, Equality (5) is a result from the fact that $\frac{\overline{p}_\alpha - x}{\lambda_\alpha} \in P \Leftrightarrow x \in P_1$.

With a similar strategy we can find $p(p_2|p_+ = \overline{p})$:

$$
\begin{aligned}
p(p_2|p_+ = \overline{p}) &= p(p_+ = \overline{p}|p_2) \cdot \frac{p(p_2)}{p(p_+ = \overline{p})} \\
&= p(p_1 = \overline{p}_\alpha - \lambda_\alpha \cdot p_2|p_2) \cdot \frac{p(p_2)}{\int_x p\left(p_1 = x, p_2 = \frac{\overline{p}_\alpha - x}{\lambda_\alpha}\right) dx} \\
&= p(p_1 = \overline{p}_\alpha - \lambda_\alpha \cdot p_2|p_2) \cdot \frac{p(p_2)}{\int_x p\left(p_1 = x, p_2 = \frac{\overline{p}_\alpha - x}{\lambda_\alpha}\right) dx} \qquad , p_2 \in P_2 \\
&\overset{(1)}{=} p(p_1 = \overline{p}_\alpha - \lambda_\alpha \cdot p_2|p_2) \cdot \frac{p(p_2)}{\int_x p\left(p_1 = x, p_2 = \frac{\overline{p}_\alpha - x}{\lambda_\alpha}\right) dx} \qquad , p_1 \in P_1 \\
&= p(p_1 = \overline{p}_\alpha - \lambda_\alpha \cdot p_2) \cdot \frac{p(p_2)}{\int_x p\left(p_1 = x, p_2 = \frac{\overline{p}_\alpha - x}{\lambda_\alpha}\right) dx} \\
&= p(p_1 = \overline{p}_\alpha - \lambda_\alpha \cdot p_2) \cdot \frac{\frac{1}{m}}{\int_x p\left(p_1 = x|p_2 = \frac{\overline{p}_\alpha - x}{\lambda_\alpha}\right) \cdot p\left(p_2 = \frac{\overline{p}_\alpha - x}{\lambda_\alpha}\right) dx} \\
&= p(p_1 = \overline{p}_\alpha - \lambda_\alpha \cdot p_2) \cdot \frac{\frac{1}{m}}{\int_{x \in P_1} p(p_1 = x) \cdot \frac{1}{m} dx + \int_{x \notin P_1} 0 dx} \\
&= \frac{p(p_1 = \overline{p}_\alpha - \lambda_\alpha \cdot p_2)}{\int_{x \in P_1} p(p_1 = x) dx} \\
&= p(p_1 = \overline{p}_\alpha - \lambda_\alpha \cdot p_2|p_1 \in P_1) \\
&\overset{(2)}{=} p(p_1 = \overline{p}_\alpha - \lambda_\alpha \cdot p_2|p_2 \in P_2) \\
p(p_2|p_+ = \overline{p}) &= p(p_1 = \overline{p}_\alpha - \lambda_\alpha \cdot p_2|p_2 \in P_2)
\end{aligned}
$$

where Equality (1) and (2) result from the fact that $p_1 \in P_1 \Leftrightarrow p_2 \in P_2$, given a specific value of $p_+ = \overline{p}$. This completes this proof. $\qquad \square$

## D.2 LEMMA 2. EXTREME MIXTURES LEMMA

**Lemma 2. *Extreme mixtures lemma.*** *Consider two independent arbitrary probability distributions $p_1$ and $p_2$. Given only a third probability distribution $p_+ = \overline{p}$ composed of the weighted mixture of the two, and for a small $\lambda \approx 0$, we can find a small residual hyperplane $P_1$, which tends to $\{\overline{p}\}$.*

$$P_1 \approx \{p_r = \overline{p} - \lambda \cdot p, \forall p \in P \mid \lambda \cdot p \le \overline{p}\} \quad \lambda \approx 0 \tag{9}$$

*We can also find a very large residual hyperplane $P_2$ for $p_2$, which tends to:*

$$\lim_{\lambda \to 0} P_2 = \{p, \forall p \in P \mid supp(p) \subseteq supp(\overline{p})\} \tag{10}$$

*where $supp(\cdot)$ is the support of a probability distribution.*

*Proof.* In this proof, we start with the arbitrary residual hyperplanes $P_r$ and find restrictions in the limits of $\lambda \to 0$ and $\lambda \to 1$. For a $\beta \approx 0$:

$$
\begin{aligned}
\lim_{\beta \to 0} P_r &= \lim_{\beta \to 0}\{p_r = \tfrac{\overline{p} - \beta \cdot p}{1 - \beta}, \forall p \in P \mid \beta \cdot p \le \overline{p}\} \\
&= \lim_{\beta \to 0}\{p_r = \overline{p} - \beta \cdot p, \forall p \in P \mid \beta \cdot p \le \overline{p}\} \\
&= \{\overline{p}\} \\
P_r &\approx \{p_r = \overline{p} - \beta \cdot p, \forall p \in P \mid \beta \cdot p \le \overline{p}\} & \beta \approx 0 \\
P_1 &\approx \{p_r = \overline{p} - \lambda \cdot p, \forall p \in P \mid \lambda \cdot p \le \overline{p}\} & \lambda \approx 0 \\
P_2 &\approx \{p_r = \overline{p} - (1 - \lambda) \cdot p, \forall p \in P \mid (1 - \lambda) \cdot p \le \overline{p}\} & \lambda \approx 1 \therefore \beta \approx 0
\end{aligned}
$$

For a $\beta \approx 1$ we start with the other definition of $P_r$:

$$
\begin{aligned}
\lim_{\beta \to 1} P_r &= \lim_{\beta \to 1} \{p_r, \forall p_r \in P \mid (1 - \beta) \cdot p_r \leq \bar{p}\} \\
&= \lim_{\beta \to 1} \{p_r, \forall p_r \in P \mid \mathrm{supp}(p_r) \subseteq \mathrm{supp}(\bar{p}), (1 - \beta) \cdot p_r \leq \bar{p}\} \\
&= \{p_r, \forall p_r \in P \mid \mathrm{supp}(p_r) \subseteq \mathrm{supp}(\bar{p})\} \\
P_r &\approx \{p_r, \forall p_r \in P \mid \mathrm{supp}(p_r) \subseteq \mathrm{supp}(\bar{p})\} & \beta \approx 1 \\
P_1 &\approx \{p_r, \forall p_r \in P \mid \mathrm{supp}(p_r) \subseteq \mathrm{supp}(\bar{p})\} & \lambda \approx 1 \\
P_2 &\approx \{p_r, \forall p_r \in P \mid \mathrm{supp}(p_r) \subseteq \mathrm{supp}(\bar{p})\} & \lambda \approx 0 \therefore \beta \approx 1
\end{aligned}
$$

This finishes this proof. $\qquad\square$

### D.3 THEOREM 1. NO FREE ANOMALY THEOREM

**Theorem 1.** *Consider two independent arbitrary probability distributions $p_{normal}$ and $p_{anom}$. For a small number of anomalies $\lambda \approx 0$, the knowledge that $p_{full} = \bar{p}$ gives us no further knowledge on the distribution of $p_{anom}$:*

$$
p(p_{anom}|p_{full} = \bar{p}) \approx Uniform(P_2), \quad \lambda \approx 0
$$

*Proof.* Consider in Lemmas 1 and 2 that $p_2 = p_{anom} \sim Uniform(P)$. We then have that, for a small value of $\lambda \approx 0$:

$$
\begin{aligned}
p(p_2|p_+ = \bar{p}_\alpha) &= p(p_1 = \bar{p}_\alpha - \lambda_\alpha \cdot p_2 | p_2 \in P_2) \\
&\approx p(p_1 = \bar{p}_\alpha | p_2 \in P_2) \\
&= Uniform(P_2)
\end{aligned}
$$

This finishes this proof. $\qquad\square$

## E   FURTHER PROOFS

In this section, we prove upper and lower bounds on the maximum distance a probability distribution $p_1$ can be from $p_+$, based on the value of $\lambda$. This can be directly applied to $p_{normal}$ for small values of $\lambda$ and to $p_{anom}$ for large ones.

**Theorem 2.** *Upper Bound on Mixture Probability Distance For two independent arbitrary probability distributions $p_1$ and $p_2$, given only a third probability distribution $p_+$ composed of the weighted mixture of the two:*

$$
p_+ = (1 - \lambda) \cdot p_1 + \lambda \cdot p_2
$$

*We have an upper bound on the distance measures $\delta(p_+, p_1)$ and $||p_+ - p_1||$ given by:*

$$
\delta(p_+, p_1) \leq \sqrt{\frac{1}{2} \log \frac{1}{1 - \lambda}}
$$

$$
||p_+ - p_1|| \leq \sqrt{2 \log \frac{1}{1 - \lambda}}
$$

*which is a tight bound for $\lambda \approx 0$. In this equation $\delta(\cdot)$ is the total variation distance between two probability distributions and $|| \cdot ||$ is the $L_1$ norm.*

*Proof.* Pinsker's inequality states that if $p$ and $q$ are two probability distributions on a common measurable space $(\mathcal{A}, \mathcal{F})$:

$$
\delta(p, q) = \sup\{|p(A) - q(A)| \ : A \in \mathcal{F}\} \leq \sqrt{\frac{1}{2} \cdot D_{KL}(p||q)}
$$

$$
||p - q|| \leq \sqrt{2 \cdot D_{KL}(p||q)}
$$

where $D_{KL}(p||q)$ is the Kullback–Leibler divergence in nats. So we have that:

$$
\delta(p_+, p_1) \leq \sqrt{\frac{1}{2} \cdot D_{KL}(p_1||p_+)}
$$

and this Kullback–Leibler divergence is itself upper-bounded by:

$$
\begin{aligned}
D_{KL}\left(p_1 || p_+\right) &= \int_x \left(p_1(x) \log \frac{p_1(x)}{p_+(x)} dx\right) \\
&= \int_x \left(p_1(x) \log \frac{p_1(x)}{(1-\lambda) \cdot p_1(x) + \lambda \cdot p_2(x)} dx\right) \\
&\leq \max_{p_2} \left(\int_x \left(p_1(x) \log \frac{p_1(x)}{(1-\lambda) \cdot p_1(x) + \lambda \cdot p_2(x)} dx\right)\right)
\end{aligned}
$$

where this maximum Kullback–Leibler divergence is achieved when $p_1$ and $p_2$ are disjoint probability distributions:

$$
\begin{aligned}
D_{KL}\left(p_1 || p_+\right) &\leq \max_{p_2} \left(\int_x \left(p_1(x) \log \frac{p_1(x)}{(1-\lambda) \cdot p_1(x) + \lambda \cdot p_2(x)} dx\right)\right) \\
&\leq \int_x \left(p_1(x) \log \frac{p_1(x)}{(1-\lambda) \cdot p_1(x)} dx\right) \\
&= \int_x \left(p_1(x) \log \frac{1}{1-\lambda} dx\right) \\
&= \log \frac{1}{1-\lambda} \int_x \left(p_1(x) dx\right) \\
&= \log \frac{1}{1-\lambda}
\end{aligned}
$$

which concludes the proof that:

$$
\delta(p_+, p_1) \leq \sqrt{\frac{1}{2} \log \frac{1}{1-\lambda}}
$$

$$
||p_+ - p_1|| \leq \sqrt{2 \log \frac{1}{1-\lambda}}
$$

$\square$

**Theorem 3.** *__Lower Bound on Maximum Mixture Probability Distance__ For two independent arbitrary probability distributions $p_1$ and $p_2$, given only a third probability distribution $p_+$ composed of the weighted mixture of the two:*

$$
p_+ = (1-\lambda) \cdot p_1 + \lambda \cdot p_2
$$

*We have a lower bound on the maximum possible distance measures $\delta(p_+, p_1)$ and $||p_+ - p_1||$ for a chosen maximizing $p_1$ given by:*

$$
max_{p_1} \delta(p_+, p_1) \geq \lambda \cdot \frac{|\mathcal{A} - 1|}{|\mathcal{A}|}
$$

$$
max_{p_1} ||p_+ - p_1|| \geq 2\lambda \frac{|\mathcal{A} - 1|}{|\mathcal{A}|}
$$

*which is a tight bound for $\lambda \approx 1$, considering the maximum $L_1$ distance between two probability distributions is $2$.*

*Proof.* We can prove a lower bound on the maximized distance of a probability distribution $p_1$ from $p_+$ by expanding the distance equations:

$$
\begin{aligned}
max_{p_1} \delta(p_+, p_1) &= max_{p_1} \sup\{|p_+(A) - p_1(A)| : A \in \mathcal{F}\} \\
&= max_{p_1} \sup\{|(1-\lambda) \cdot p_1 + \lambda \cdot p_2 - p_1(A)| : A \in \mathcal{F}\} \\
&= max_{p_1} \sup\{|\lambda \cdot p_2(A) - \lambda \cdot p_1(A)| : A \in \mathcal{F}\} \\
&= \lambda \cdot max_{p_1} \sup\{|p_2(A) - p_1(A)| : A \in \mathcal{F}\} \\
&\overset{(a)}{\geq} \lambda \cdot max_{p_1} min_{p_2} \sup\{|p_2(A) - p_1(A)| : A \in \mathcal{F}\} \\
&\overset{(b)}{=} \lambda \cdot max_{p_1} \sup\{|Uniform(\mathcal{A}) - p_1(A)| : A \in \mathcal{F}\} \\
&\overset{(c)}{=} \lambda \cdot \sup\{|Uniform(\mathcal{A}) - \delta(A)| : A \in \mathcal{F}\} \\
&= \lambda \cdot \frac{|\mathcal{A}-1|}{|\mathcal{A}|} \\
max_{p_1} \delta(p_+, p_1) &\geq \lambda \cdot \frac{|\mathcal{A}-1|}{|\mathcal{A}|}
\end{aligned}
$$

where in (a) we lower bound based on the probability distribution that would have the smallest possible superior distance to a later maximized probability distribution $p_1$. This probability distribution $p_1$ can always maximize its superior distance to $p_2$ by:

$$p_1(a) = \begin{cases} 1 & , if\ a = \mathrm{argmin}_x(p_2(x)) \\ 0 & , else \end{cases}$$

In (b) we choose the uniform distribution as the one that would reduce this superior distance and in (c) we set $p_1(a) = 1$ for a random $a$, since $p_2$ is uniform. With a similar strategy we find:

$$max_{p_1}||p_+ - p_1|| \geq 2\lambda\frac{|\mathcal{A} - 1|}{|\mathcal{A}|}$$

This concludes this proof. $\square$

