# OpenReview forum: "UaiNets: From Unsupervised to Active Deep Anomaly Detection"
_ICLR.cc/2019/Conference_

### Official Review · AnonReviewer3 · 2018-11-02
**Active anomaly detection technique employing existing approaches and lacking appropriate literature review**

**Rating:** 3
**Confidence:** 4

**Review:**

(Since the reviewer was unclear about the OpenReview process, this review was earlier posted as public comment)

Most claims of novelty can be clearly refuted such as the first sentence of the abstract "...This work presents a new approach to active anomaly detection..." and the paper does not give due credit to existing work. Current research such as Das et al. which is the most relevant has been deliberately not introduced upfront with other works (because it shows lack of the present paper's novelty) and instead deferred to later sections. The onus of a thorough literature review and laying down a proper context is on the authors, not the reviewers. Detailed comments are below.

      1. Related Works: "...active anomaly detection remains an under-explored approach to this problem..."
          - Active learning in anomaly detection is well-researched (AI2, etc.). See related works section in Das et al. 2016 and:
            - K. Veeramachaneni, I. Arnaldo, A. Cuesta-Infante, V. Korrapati, C. Bassias, and K. Li, "Ai2: Training a big data machine to defend," International Conference on Big Data Security, 2016.

      2. "To deal with the cold start problem, for the first 10 calls of select_top...":
          - No principled approach to deal with cold start and one-sided labels (i.e., the ability to use labels when instances from only one class are labeled.)

      3. Many arbitrary hyper parameters as compared to simpler techniques:
          - The number of layers, nodes in hidden layers.
            - The number of instances (k) per iteration
            - The number of pretraining iterations
            - The number of times the network is retrained (100) after each labeling call
            - Dealing with cold start (10 labeling iterations of 10 labels each, i.e. 100 labels).

      4. The paper mentions that s(x) might not be differentiable. However, the sigmoid form of s(x) is differentiable.

      5. Does not acknowledge the well-known result that mixture models are unidentifiable. The math in the paper is mostly redundant. Some references:
          - Identifiability  Of  Nonparametric  Mixture  Models And  Bayes  Optimal  Clustering (pradeepr/arxiv npmix v.pdf)" target="_blank" rel="nofollow">https://www.cs.cmu.edu/ pradeepr/arxiv npmix v.pdf)
          - Semiparametric estimation of a two-component mixture model by Bordes, L., Kojadinovic, I., and Vandekerkhove, P., Annals of Statistics, 2006 (https://arxiv.org/pdf/math/0607812.pdf)
          - Inference for mixtures of symmetric distributions by David R. Hunter, Shaoli Wang, Thomas P. Hettmansperger, Annals of Statistics, 2007 (https://arxiv.org/pdf/0708.0499.pdf)
          - Inference on Mixtures Under Tail Restrictions by K. Jochmans, M. Henry, and B. Salanie, Econometric Theory, 2017 (http://econ.sciences-po.fr/sites/default/files/file/Inference.pdf)

      6. Does not acknowledge existing work that adds classifier over unsupervised detectors (such as AI2). This is very common.
        - This is another linear model (logistic) on top of transformed features. The difference is that the transformed features are from a neural network and optimization can be performed in a joint fashion. The novelty is marginal.

      7. While the paper argues that a prior needs to be assumed, it does not use any in the algorithm. There seems to be a disconnect. It also does not acknowledge that AAD (LODA/Tree) does use a prior. Priors for anomaly proportions in unsupervised algorithms are well-known (most AD algos support that such as OC-SVM, Isolation Forest, LOF, etc.).

      8. Does not compare against current state-of-the-art Tree-based AAD
          - Incorporating Expert Feedback into Tree-based Anomaly Detection by Das et al., KDD, 2017.

      9. The 'Generalized' in the title is incorrect and misleading. This is specific to deep-networks. Stacking supervised classifiers on unsupervised detectors is very common. See comments on related works.

      10. Does not propose any other query strategies than greedily selecting top.

      11. Question: Does this support streaming?

---

> ### Author Response · Authors · 2018-11-22
> **Thank you for your very thorough and detailed review! And further clarifications.**
>
> We would like to again thank you for your very thorough and detailed review.
> We tried to incorporate all the feedback into the manuscript, starting by changing its title to 'UaiNets: From Unsupervised to Active Deep Anomaly Detection'.
> We believe this title better represents the main point of this work: translating unsupervised deep anomaly detection models to active ones.
> We also tried to change the verbiage in the paper to make this clearer.
>
> We have already addressed most of the points raised in the commentary bellow (before rebuttal period), but now that we incorporated it into the manuscript would like to readdress a few.
>
> 1. Related Works: "...active anomaly detection remains an under-explored approach to this problem..."
> -> We still believe it is under explored, although we made it clear that there are some very interesting prior work on this.
>
> 4. The paper mentions that s(x) might not be differentiable. However, the sigmoid form of s(x) is differentiable.
> -> We ran it allowing gradients through s(x) and the network improves on most datasets.
> -- But, since the underlying models might have non differentiable s(x), we kept the old results in the paper. We believe they are more representative.
>
> 5. Does not acknowledge the well-known result that mixture models are unidentifiable. The math in the paper is mostly redundant. Some references:
> -> We added a short acknowledgement in Section 2.1.
>
> 6. Does not acknowledge existing work that adds classifier over unsupervised detectors (such as AI2). This is very common.
> -> We read AI2 carefully and, if we understood correctly, it does not add a classifier over unsupervised detectors.
> -- It uses both a supervised classifier (random forest) and an unsupervised ensemble in a parallel manner to find anomalies.
> -- It trains both the unsupervised and supervised using the same set of features M, although the supervised model is only trained on the already labeled instances.
> -- So the only other work that adds classifier over unsupervised detectors would be LODA-AAD and Tree-AAD, which we address as really important prior work.
> -- We do it differently than they do though.
>
> -- To the best of our knowledge ours is the first work which applies deep learning to active anomaly detection.
> -- We believe this is also the first work to approach active anomaly detection in an end-to-end manner.
> ---   Other work, such as LODA-AAD and Tree-AAD have a phase where they train their underline algorithms, and another which uses labeled instances to learns weights to change these underlying results.
> ---   At the same time, AI^2 learns two separate models, an unsupervised and a supervised one, each with its advantages.
> ---   Ours uses a composite loss function composed of the underlying model's loss added to the UAI layer one (binary cross entropy only applied to labeled instances).
> ---   We tried to make this clearer in the text.
> -- Figures 9 and 10 in Appendix C.1 show why learning representations end-to-end is important.
>
> 8. Does not compare against current state-of-the-art Tree-based AAD
> -> Added comparison
>
> 9. The 'Generalized' in the title is incorrect and misleading. This is specific to deep-networks. Stacking supervised classifiers on unsupervised detectors is very common. See comments on related works.
> -> We changed the title.
>
> 10. Does not propose any other query strategies than greedily selecting top.
> -> We believe this would be a very interesting research topic, one we want to study, but it deserves a paper on its own.
> -- It would be too dense a topic to cram it in here, so we followed the same strategy as prior work when doing this.
> -- This paper already had 25 pages and we could not address this issue here.

---

> > ### Comment · AnonReviewer3 · 2018-11-24
> > **Addresses some of the review comments**
> >
> > Dear Authors, I appreciate your addressing some of the review comments. However, some major issues with the paper remain:
> >
> > 1. Simply plugging deep-learning with active learning (for anomaly detection) is not a significant contribution.
> >
> > 2. The theory in the paper is completely redundant and its implications are already well-known and well-appreciated. There is no need to introduce a superficial 'no free anomaly' theorem. It is independent of the algorithm and hence seems quite out-of-place.
> >
> > 3. It is correct that the paper lacks space for some critical aspects. However, this is a problem arising out of the paper's organization. For instance, the theory and the results on the synthetic data such as Figure 3 are not relevant and uninteresting. You could remove all these and try addressing the more important concerns which are of significance to anomaly detection and which possibly extend active learning with deep learning into new territory. The current approach is merely plugging in one black box over other older ones.

---

### Official Review · AnonReviewer2 · 2018-11-03
**Interesting paper that can be significantly improved by a better organization.**

**Rating:** 5
**Confidence:** 2

**Review:**

This is an interesting paper on a topic with real-world application: anomaly detection.

The paper's organization is, at times quite confusing:
- the introduction is unusually short, with a 1st paragraph virtually unreadable due to the abuse of citations. Two additional paragraphs, covering in an intuitive manner both the proposed approach & the main results, would dramatically improve the paper's readability
- section 2.1 starts quite abruptly with he two Lemmas 7 and Theorem 3 (which, in fact, is Theorem 1). This section would probably read a lot better without the two Lemmas, as the authors only refer to the main result in the Theorem. The second, intuitive part of 2.1 is extremely helpful.
- it is unclear why the authors have applied the approach in "4.3" only to a single dataset, rather than all the 11 datasets

Other comments:
- please change the color schemes for Figures 3 & 4, where the red/orange (Fig 3) and various blues (Fig 4) are difficult to distinguish
- bottom of page 3: "are rare as expected" --> "are as rare as expected"

---

> ### Author Response · Authors · 2018-11-22
> **Thank you for your comments!**
>
> Thank you for your comments. We tried to change the verbiage in the paper to make it clearer.
> We address each point bellow:
>
> the introduction is unusually short, with a 1st paragraph virtually unreadable due to the abuse of citations. Two additional paragraphs, covering in an intuitive manner both the proposed approach & the main results, would dramatically improve the paper's readability
> ->We added a more detailed explanation of the architecture into the introduction and moved the schematic in Figure 1 here as well to try to exemplify it.
>
> section 2.1 starts quite abruptly with he two Lemmas 7 and Theorem 3 (which, in fact, is Theorem 1). This section would probably read a lot better without the two Lemmas, as the authors only refer to the main result in the Theorem. The second, intuitive part of 2.1 is extremely helpful.
> -> We also moved the lemmas to the appendix, trying to make this clearer.
>
> it is unclear why the authors have applied the approach in "4.3" only to a single dataset, rather than all the 11 datasets
> -> There are only two datasets that already have a test set with new classes of anomalies: KDDCUP and KDDCUP-rev.
> -- We ran only on KDDCUP-rev because the LODA-AAD takes too long on KDDCUP for this experiment.
> ---     There are 311029 test instances in KDDCUP and 67908 in KDDCUP-rev
> ---     And there are 494021 train instances in KDDCUP and 121597 in KDDCUP-rev
> ---     The analysis already took a couple of days on KDDCUP-rev.
> ---     It ran 16 times slower in KDDCUP (4 times less expert feedback and 4 times less test data)
> -- For each expert feedback iteration, we ran the model in the full test set.
>
> please change the color schemes for Figures 3 & 4, where the red/orange (Fig 3) and various blues (Fig 4) are difficult to distinguish
> -> There was a problem with the hard drive where results were saved and we lost them.
> -- We will rerun experiments and try to fix this.
>
> bottom of page 3: "are rare as expected" --> "are as rare as expected"
> -> Changed it
>
> We hope this clears any confusing points the paper might have made. If it doesn't, we would be pleased to answer any other questions/suggestions.

---

### Official Review · AnonReviewer1 · 2018-11-07
**An interesting problem : active learning for anomaly detection; method suffering from a lack of novelty; questions about experiments**

**Rating:** 4
**Confidence:** 4

**Review:**

The paper provided a convincing and intuitive motivation regarding the need for active learning in unsupervised anomaly detection.
However the proposed approach of requesting expert feedback for the top ranked anomalies is straightforward and unsurprising, given past work on active learning.
The experiments on synthetic data are also unsurprising. Moreover these are based on a questionable premise: the instances that are "hard" to classify are treated as anomalies. This is not very realistic.
Regarding the real data experiments: In Table 1 the results for DAE_uai are based on which budget b?  How does the result vary with b?

---

> ### Author Response · Authors · 2018-11-22
> **Thank you for the feedback!**
>
> We would like to start by thanking you for the feedback, we tried to incorporate it in the manuscript.
> We also address each of your points bellow:
>
> The paper provided a convincing and intuitive motivation regarding the need for active learning in unsupervised anomaly detection.
> -> Thank you!
>
> However the proposed approach of requesting expert feedback for the top ranked anomalies is straightforward and unsurprising, given past work on active learning.
> -> Could you elaborate more in which way it is unsurprising?
> -- We tried to make clearer that the main contribution of our work is not that it uses active learning, but how it does so.
> -- To the best of our knowledge this is the first work which applies deep learning to active anomaly detection.
> -- We believe this is also the first work to approach active anomaly detection in an end-to-end manner.
> ---   Other work, such as LODA-AAD and Tree-AAD have a phase where they train their underlying algorithms, and another which uses labeled instances to learns weights to change these underlying results.
> ---   At the same time, AI^2 learns two separate models, an unsupervised and a supervised one, each with its advantages.
> ---   Ours uses a composite loss function composed of the underlying model's one added to the UAI layer loss (binary cross entropy only applied to labeled instances).
> ---   We tried to make this clearer in the text.
> -- Figures 9 and 10 in Appendix C.1 show why learning representations end-to-end is important.
>
> The experiments on synthetic data are also unsurprising. Moreover these are based on a questionable premise: the instances that are "hard" to classify are treated as anomalies. This is not very realistic.
> -> This experiments' results may be unsurprising.
> -- Nonetheless, we believe they are interesting to test the assumption that our model can indeed deal with different 'types' of anomalies, even when the underlying model can't.
> -- Presenting empirical results for this.
> -- Furthermore, we believe that even if one of the anomaly types is not realistic they serve their purpose, of testing the robustness of the UaiNets compared to the underlying models.
>
> Regarding the real data experiments: In Table 1 the results for DAE_uai are based on which budget b?  How does the result vary with b?
> -> For all these experiments we used b as the number of anomalies in the dataset.
> -- The model is robust to the choice of b, this can be seen in appendix C.2.
> -- The larger the b the better the algorithm becomes, but it can usually already learn pretty well even with few labels.
> -- Figure 11 shows that for KDDCUP, Thyroid, Arrhythmia and KDDCUP-Rev the algorithm improves significantly with a few examples.

---

### Public Comment · (anonymous) · 2018-10-11
**Ignores existing work in statistics and semi-supervised anomaly detection, and does not make principled effort to overcome practical challenges like cold start and one-sided labels.**

This work starts making clearly refuted claims of novelty right from the first sentence of the abstract "...This work presents a new approach to active anomaly detection..." and does not give due credit to existing work. Current research such as Das et al. which is the most relevant has been deliberately not introduced upfront with other works (because it shows lack of the present paper's novelty) and instead deferred to later sections. The onus of a thorough literature review and laying down a proper context is on the authors, not the reviewers. Detailed comments are below.

1. Related Works: "...active anomaly detection remains an under-explored approach to this problem..."
    - Active learning in anomaly detection is well-researched (AI2, etc.). See related works section in Das et al. 2016 and:
      - K. Veeramachaneni, I. Arnaldo, A. Cuesta-Infante, V. Korrapati, C. Bassias, and K. Li, "Ai2: Training a big data machine to defend," International Conference on Big Data Security, 2016.

2. "To deal with the cold start problem, for the first 10 calls of select_top...":
    - No principled approach to deal with cold start and one-sided labels (i.e., the ability to use labels when instances from only one class are labeled.)

3. Many arbitrary hyper parameters as compared to simpler techniques:
    - The number of layers, nodes in hidden layers.
      - The number of instances (k) per iteration
      - The number of pretraining iterations
      - The number of times the network is retrained (100) after each labeling call
      - Dealing with cold start (10 labeling iterations of 10 labels each, i.e. 100 labels).

4. The paper mentions that s(x) might not be differentiable. However, the sigmoid form of s(x) is differentiable.

5. Does not acknowledge the well-known result that mixture models are unidentifiable. The math in the paper is mostly redundant. Some references:
    - Identifiability  Of  Nonparametric  Mixture  Models And  Bayes  Optimal  Clustering (https://www.cs.cmu.edu/~pradeepr/arxiv_npmix_v2.pdf)
    - Semiparametric estimation of a two-component mixture model by Bordes, L., Kojadinovic, I., and Vandekerkhove, P., Annals of Statistics, 2006 (https://arxiv.org/pdf/math/0607812.pdf)
    - Inference for mixtures of symmetric distributions by David R. Hunter, Shaoli Wang, Thomas P. Hettmansperger, Annals of Statistics, 2007 (https://arxiv.org/pdf/0708.0499.pdf)
    - Inference on Mixtures Under Tail Restrictions by K. Jochmans, M. Henry, and B. Salanie, Econometric Theory, 2017 (http://econ.sciences-po.fr/sites/default/files/file/Inference.pdf)

6. Does not acknowledge existing work that adds classifier over unsupervised detectors (such as AI2). This is very common.
  - This is another linear model (logistic) on top of transformed features. The difference is that the transformed features are from a neural network and optimization can be performed in a joint fashion. The novelty is marginal.

7. While the paper argues that a prior needs to be assumed, it does not use any in the algorithm. There seems to be a disconnect. It also does not acknowledge that AAD (LODA/Tree) does use a prior. Priors for anomaly proportions in unsupervised algorithms are well-known (most AD algos support that such as OC-SVM, Isolation Forest, LOF, etc.).

8. Does not compare against current state-of-the-art Tree-based AAD
    - Incorporating Expert Feedback into Tree-based Anomaly Detection by Das et al., KDD, 2017.

9. The 'Generalized' in the title is incorrect and misleading. This is specific to deep-networks. Stacking supervised classifiers on unsupervised detectors is very common. See comments on related works.

10. Does not propose any other query strategies than greedily selecting top.

11. Question: Does this support streaming?

---

> ### Author Response · Authors · 2018-11-01
> **Thank you for the feedback.**
>
> Thank you for your comment and we appreciate the feedback, we will incorporate suggestions in our manuscript. In this work we present new methods based on the proposed new architectures (UaiNets), which we see as a new approach to active anomaly detection. This might be better phrased as "This work presents new active anomaly detection methods". And we do give credit to Das et al. stating " The most similar prior work to ours in this setting is (Das et al., 2016), which proposed an algorithm that can be employed on top of any ensemble methods based on random projections.", but we should have mentioned it in Section 3.1 when we describe our approach and we will fix this during the rebuttal period. Nonetheless this was not ill intended or deliberate.
> We address each of your detailed comments bellow:
>
> 1. We still believe it is an under-explored approach to this problem. In the well known (Chandola et al. 2009) survey, they don't mention active learning at all. Only citing (Abe et al. 2006) as supervised anomaly detection. (Das et al. 2016) only has 12 citations and (Das et al. 2017) has only one self-citation. These are some really interesting works in this area, but we believe if this was a well-researched topic they would have more recognition (assuming citations can be used as a measure for recognition).
>    Nonetheless, we should indeed have cited (Veeramachaneni et al. 2016) and (Das et al. 2017). We will add it during the rebuttal phase.
> - Chandola, V., Banerjee, A. and Kumar, V., 2009. Anomaly detection: A survey. ACM computing surveys (CSUR), 41(3), p.15.
> - Abe, N., Zadrozny, B. and Langford, J., 2006, August. Outlier detection by active learning. In Proceedings of the 12th ACM SIGKDD international conference on Knowledge discovery and data mining (pp. 504-509). ACM.
> - Das, S., Wong, W.K., Dietterich, T., Fern, A. and Emmott, A., 2016, December. Incorporating expert feedback into active anomaly discovery. In Data Mining (ICDM), 2016 IEEE 16th International Conference on (pp. 853-858). IEEE.
> - Das, S., Wong, W.K., Fern, A., Dietterich, T.G. and Siddiqui, M.A., 2017. Incorporating Feedback into Tree-based Anomaly Detection. arXiv preprint arXiv:1708.09441.
> - Veeramachaneni, K., Arnaldo, I., Korrapati, V., Bassias, C. and Li, K., 2016, April. AI^ 2: training a big data machine to defend. In Big Data Security on Cloud (BigDataSecurity), IEEE International Conference on High Performance and Smart Computing (HPSC), and IEEE International Conference on Intelligent Data and Security (IDS), 2016 IEEE 2nd International Conference on (pp. 49-54). IEEE.
>
> 2. Our architecture can be built on top of state-of-the-art unsupervised anomaly detection models, so using them during our models cold start is a good option. it gives state of the art anomaly detection in these first steps.
>    We state in the paper though that an interesting future work would be "using the UAI layers confidence in its output to dynamically choose between either directly using its scores, or using the underlying unsupervised model’s anomaly score to choose which instances to audit next".
>    This is not straight forward though, since confidence scores from deep learning architectures are usually unregulated.
>
> 3. Our model has several hyper parameters, but we show through our experiments that the network can produce good results to all analyzed datasets with the same choice of hyper parameters.
>    We only change k when dealing with datasets with few anomalies to give the model the chance to further interact with labels.
>    Our algorithm is robust to k. For k ∈ {5; 10; 20; 30; 40; 50; 100}, using KDDCUP-rev, we get F1 scores of {0.90; 0.91; 0.90; 0.90; 0.91; 0.91; 0.91}, respectively, with no statistical difference between them (p < 0.1).
>    The choice of k is left to the user, since it might depend on their business model.
>    A large company with several experts might want to parallelize the models feedback and get more instances per iteration with the model.
>
> 4. Both base models used have differentiable s(x) (squared error in DAE and sigmoid in classifier), but we wanted to build an architecture which could (potentially) be applied to different deep learning models in the future. Since this models might have non differentiable s(x) we didn't allow gradients to flow through it in our experiments.
>    we didn't test this, but we believe we might actually see better results if we allowed gradients through s(x).
>
> 5. We believe our results expand on the unidentifiability of mixture models, showing that in this case *all* possible options are equally unidentifiable.
>    Nonetheless we should have cited these results and will mention and compare to them during rebuttal phase.

---

> > ### Author Response · Authors · 2018-11-01
> > **Thank you for the feedback. - Continuation**
> >
> > 6. We do not think the novelty is marginal.
> >    Deep Learning architectures excel exactly as feature extractors, being great for learning representations.
> >    Besides, we show through the experiments in Appendix C.1 that end to end learning helps the base architecture learn better feature representations for anomaly detection.
> >
> > 7. We do not argue that a prior needs to be assumed for all cases (although the no free lunch theorem does). We only argue that unsupervised anomaly detection needs one.
> >    Supervised algorithms have, in general, presented good priors for most supervised learning problems, and the UAI layer learns in a supervised (active) way.
> >    We also show in Section 4.1 that although unsupervised active learning have to trade off accuracy in a setting for another, active algorithms are robust to their choice and can give good results in all analyzed settings.
> >
> > 8. We should have compared to them. Here are the results:
> > Tree-AAD      0.89* 0.29 0.86 0.50 0.32 0.53 0.69 0.76 0.94 0.59 0.92
> > DAE_uai        0.94   0.47 0.57 0.91 0.33 0.55 0.66 0.64 0.86 0.60 0.93
> > In order: KDDCUP, Arrhythmia, Thyroid, KDDCUP-Rev, Yeast, Abalone, CTG, Credit Card, Covtype, MMG, Shuttle
> > * to to run Tree-AAD on KDDCUP we needed to limit its memory about the anomalies it had already learned, forgetting the oldest ones. This reduced its runtime complexity from O(b^2) to O(b) in our tests, where b is the budget limit for the anomaly detection task.
> >
> > 9. We can see how it might be misleading and will consider changing the title.
> >
> > 10. Greedily selecting top is a good strategy in practical settings and (Das et al. 2016) and (Das et al. 2017) also use it.
> >     In practical scenarios we want to have the most anomalies for a given budget, so selecting the most anomalous instance at a time is a useful strategy.
> >     Also, if we select a non anomalous instance, we will use it to correct our probability distribution, improving our results in the next iteration.
> >     Finally, since anomaly detection is already a highly imbalanced setting, we might not get anomalous instances even when picking top anomalous results, so actively searching them might be a good choice.
> >
> > 11. What do you mean by streaming?
> >     Section 4.3 shows a setting when we have new anomalous instances arriving and we want to detect anomalies in it.
> >     If you wanted to run it on streaming data you would need to revisit the previously labeled instances every one in a while to keep training on them, while continuing training the base model on the streamed data.

---

### Meta-Review · Area_Chair1 · 2018-12-15
**Lacks demonstrated research contribution beyond past work**

**Confidence:** 5
**Recommendation:** Reject

**Metareview:**

Following the unanimous vote of the reviewers, this paper is not ready for publication at ICLR. The most significant concern raised is that there does not seem to be an adequate research contribution. Moreover, unsubstantiated claims of novelty do not adequately discuss or compare to past work.